

# The role of precipitation for high-magnitude flood generation in a large mountainous catchment (upper Rhône River, NW European Alps)

Florian Raymond[1], Bruno Wilhelm[1], and Sandrine Anquetin[1]

[1]Institute for Geosciences and Environmental Research, University Grenoble Alpes, CNRS, IRD, Grenoble-INP*, Grenoble, France

*Institute of Engineering Univ. Grenoble Alpes

*Correspondence to:* Bruno Wilhelm (bruno.wilhelm@univ-grenoble-alpes.fr)

**Abstract.** High-impact climate events such as floods are highly destructive natural hazards causing widespread impacts on socio-ecosystems. However, processes leading to such events are still poorly understood, which limiting reliable prediction. This study takes advantage of centennial-long discharge series (1923-2010) and meteorological reanalysis (ERA-20C) to study processes generating the high-magnitude flood events (i.e. above the percentile 99.9) of the upper Rhône River (NW European Alps). A particular focus is paid to the role of precipitation on the flood generation to explore in what extent such events could be explained by only atmospheric variables. A flood typology is thus established using a hierarchical clustering analysis and three variables: long (8-day) and short (2-day) precipitation accumulations as well as an index characterizing the amplitude of the discharge increase during the 7 days prior to the flood day. The typology result in four classes, of which two are directly linked to precipitation. One results from heavy precipitation over two days (similar to "short-rain floods" in the literature) and the other one from a combination of short and long intense precipitation sequences (similar to "long-rain floods"). The two other types of floods cannot be explained by precipitation only, most probably involving ice and snow melting. The four events of highest magnitude (>20 year return period) are of various types but are all triggered by heavy precipitation during the days preceding the floods. The role of the precipitation accumulations progressively decreases when considering floods of weaker magnitude, suggesting a higher diversity of processes involved in the generation of e.g. annual flooding. Our results highlight the needs to better understand the atmospheric processes leading to heavy precipitation accumulation since this would allow a better understanding of past and future trends of extreme flood events.

## 1 Introduction

From the 1980s, the number of reported floods associated with important losses has considerably increased (Kundzewicz et al., 2014). In the context of climate change, frequency and magnitude of these events are expected to change, which constitutes an increasingly relevant issue for the scientific community and the stakeholders. However, processes leading to such events are still poorly understood, e.g. limiting reliable prediction (Kundzewicz et al., 2016). This partly results from various interplays between meteorological and hydrological processes, in which interdependent variables are included at multiples space and time-scales (Merz et al., 2014). In mountainous areas, the enhanced variability of many parameters (e.g. elevation, slopes and orientations) may make such interplays even more complex. The poor understanding of the flood-generation processes is also limited by the availability of flood records at gauging stations in space and time (Hall et al., 2014; Merz et al., 2014). This is particularly true when considering rare, high-magnitude events that cause the largest impacts on socio-ecosystems.




To improve our understanding of the physical processes and the occurrence probabilities of Alpine flood events, Merz and
Blöschl (2003) used a conceptual rainfall-runoff model to analyse multiple processes associated with floods such as rainfall
regime, air temperature, potential evapotranspiration, state of the catchment and catchment characteristics. Considered flood
events were the maximum annual flood peaks (from 1971 to 1997) of 490 small Austrian catchments (sizes ranging from 3
to 30000 km²), which were grouped into five process-based flood types: (i) the flash floods, occurring mainly in small
catchments due to short (half day maximum), high-intensity rainfalls of convective origin; (ii) the short-rain floods triggered
by intense rainfalls lasting one day maximum; (iii) the long-rain floods caused by rainfall episodes lasting several days
(including low intensity rainfall); (iv) the rain-on-snow floods due to precipitation falling on an existing snow cover and (v)
the snowmelt floods caused by snowmelt during warm fair weather. With the same objective, Sikorska et al. (2015) applied
a peak-over-threshold approach (POT) method on 30-year-long discharge series and detected 5 to 10 flood events per year on
9 small Swiss catchments (catchment sizes ranging from 22 to 939 km²). To classify the 2002 identified flood events, several
dynamic and static indices based on hourly and daily data series of meteorological (precipitation), cryospheric (snow and ice
cover, snowmelt) and hydrological (discharge) observations as well as catchment characteristics (catchment wetness, geology,
topography and land use) were used. This resulted in the same five flood types of Merz and Blöschl (2003), plus a sixth one
called glacier-melt floods caused by high glacial melting due to air warming. More recently, Brunner et al. (2017) also used
the peak-over-threshold approach (POT) to study about four floods per years in 39 small and medium Swiss catchments (sizes
ranging from 20 to 1700 km²). They analysed hourly discharge data series from 17 to 53 years length and grouped the flood
events into the six flood types of Sikorska et al. (2015). This allowed to better characterise the shape of the hydrographs
associated with each of the six flood types and, thereby, improving flood risk management through a more relevant design of
hydraulic structures. Keller et al. (2018) focused on one medium Swiss catchment (1 702 km²) to create a typology based on
47 flood events detected with a POT approach applied on hourly runoff data covering the 1961-2014 period. This typology
relies on indices based on daily precipitation, temperature, snow cover and snow melt data series. This resulted in 5 flood
types that differ from the flood types identified by the previous studies since they are mainly characterised by duration and
intensity of the precipitations with two types based on long duration precipitations (with various intensities) and two types on
short precipitation duration (also with various intensities). Overall, these studies highlight i) variable combinations of
hydrological and meteorological processes for (sub-)annual flood generation mostly in small to medium mountain catchments
and ii) the large panel of hourly to daily data series (i.e discharge, precipitation, temperature, snow cover, ice cover and
catchment characteristics) necessary to properly describe these combinations of processes.

To understand processes involved in the generation of exceptional flooding, many works focused on single case studies of
recent, very detailed (e.g. Borga et al., 2007; Blöschl et al., 2013) or past, lesser informed (e.g. Ruiz-Bellet et al., 2015;
Brönniman et al., 2018; Stucki et al., 2018) events. These studies highlight the dominant roles of both precipitation
accumulation and soil saturation in the generation of such events. This is in agreement with Merz and Blöschl (2003), which
revealed a dominant role of precipitation for generation of >10-year return period events. To our knowledge, an intermediate
approach between the study of single, exceptional floods and the study of (sub-)annual floods through a process-based
typology has never been performed, while high-magnitude events are characterized by a high-impact potential on socio-
ecosystems. This might be explained by the following limitation. Studying high-magnitude flood events requires long data
series to capture a sample of flood events large enough to properly analyse processes at their origin (e.g. Brönnimann et al.,
2013) since these events occur at a much lower frequency than (sub-)annual scale. Using daily discharge data (instead of
hourly data) may overcome this difficulty as longer observation records at daily scale are then available in many regions



(Keller et al., 2018). The use of daily data, however, limits the study of processes for flood generation to large catchments,
for which the response time is of at least 1 day. Regarding meteorological data, the recent production of meteorological
reanalyses has made available a large dataset over longer periods, i.e. from 1852 (20CR, Compo et al., 2011) and 1900 (ERA-
20C, Poli et al., 2016). When data series of discharge and meteorological variables are thus available from the beginning of
the 20th century, this is, however, not the case for data series on cryosphere, i.e. data related to snow cover and snow / ice
melting.

In this context, this study aims to establish a process-based typology of high-magnitude events that occurred in a large
catchment of a mountainous area (upper Rhône River, NW European Alps) using centennial-long meteorological
(precipitation) and hydrological (discharge) datasets. Our objective is to explore in what extent the generation of high-
magnitude flood events in a large catchment can be explained by precipitation only, assuming that rain-on-snow and snow or
ice melting play thus a negligible role as observed by e.g. Merz and Blöschl (2003).

Section 2 introduces the studied area and the data used. Section 3 details the three indices used for performing the high-
magnitude flood typology. Sections 4 and 5 discuss the characteristics and relevance of each flood type.

## 94  2 Studied area and data

### 95  2.1 The upper Rhône River catchment and gauge station data

The catchment of the upper Rhône River (10 900 km²) is located in the northern French and eastern Swiss Alps (Fig. 1). The
climate influence is mainly continental with the westerlies bringing moisture from the Atlantic Ocean. At low elevations, this
results in mean annual precipitations ranging from 600 mm (in some parts of Valais, Switzerland) to 1 100 mm (Chamonix,
France). Rainy days represent 30 to 45 % with an annual maximum daily precipitation accumulation reaching 45 to 105
mm/day on average (Isotta et al., 2014). The hydrologic regime of the upper Rhône River at the gauge station of
Rhône@Bognes (Table 1) is glacio-nival with the lowest and highest daily discharges occurring respectively in December-
January (about 270 m$^3$.s$^{-1}$) and June-July (about 530 m$^3$.s$^{-1}$) for a mean daily discharge of about 359 m$^3$.s$^{-1}$. This gauge station
of Rhône@Bognes, located at Injoux-Génissiat in France, 46 km downstream to the confluence of the Arve and Rhône Rivers,
corresponds to the considered outlet of the upper Rhône catchment in this study (Fig. 1). To study the flood dynamic of the
upper Rhône River, three sub-catchments have been considered in this study and are called the Geneva, Arve and Valserine
catchments hereafter.

The Geneva catchment (8 000 km²) corresponds to the Rhône River catchment feeding Lake Geneva (Fig. 1). It is mainly
located in a Swiss high-elevation mountainous area (i.e. Valais canton), characterized by a mean and maximal altitude of 1
660 and 4 634 m a.s.l., resulting in numerous and large glaciers. For different reasons (e.g. flood protection, agricultural needs),
most of the Rhône River in the Valais has been dammed up during the 19th and the 20th centuries (Bender, 2004). In the 1950s,
7 dams have been built on many Rhône tributaries, mainly for hydroelectric production (Hingray et al., 2014). The Geneva
catchment includes Lake Geneva, the largest lake of Western Europe (580 km²), mostly fed by the Rhône River coming from
the Valais (75 % of the lake's water supply; Grandjean, 1990). At the lake outlet, the discharge has been controlled since 1884
to counter the rise in lake level that caused flooding and impacted lakefront residents. The gauge station used to evaluate the
contribution of the Geneva catchment to flood generation at Rhône@Bognes is located at the outlet of Lake Geneva in the city
of Geneva at Halle de l'Île (called Rhône@HDI hereafter, Table 1; Fig. 1). The mean daily discharge at Rhône@HDI is about





250 m$^3$.s$^{-1}$, contributing on average to 70 % of the Rhône@Bognes discharge. In the Geneva catchment, the discharge of the
Rhône River is strongly influenced by ice melting, resulting in a well-marked glacio-nival regime of Rhône@HDI with the
highest mean discharges observed between June and July (about 365 and 401 m$^3$.s$^{-1}$ on average).

The Arve catchment (1 900 km²) corresponds to a high-elevation French mountainous area, with a mean and maximal altitude
of 1 370 and 4 810 m a.s.l. (Mont Blanc, highest summit in the Alps), respectively (Fig. 1). The Mont Blanc massif that
encompasses many glaciers corresponds to the headwater catchment of the Arve River. The daily variability of the Arve
discharge over the last century is recorded at the gauge station of Bout du Monde (called Arve@BDM hereafter, Table 1 and
Fig.1), located in the city of Geneva just before the confluence with the Rhône River. The mean daily discharge at Arve@BDM
is about 79 m$^3$.s$^{-1}$ and contributes on average to 22 % of the Rhône@Bognes discharge. The discharge at Arve@BDM is
dominated by snow-melt contribution (nival regime) with the highest mean discharges observed in June (about 131 m$^3$.s$^{-1}$).

The Valserine catchment (about 1 000 km²) includes the Valserine River and smaller tributaries of the Rhône upstream the
station of Rhône@Bognes and downstream Rhône@HDI and Arve@BDM, i.e. all coming from the Jura massif (Fig. 1). No
gauge station records discharge of this catchment. Consequently, its mean daily discharge is estimated by subtracting
discharges from Rhône@HDI and Arve@BDM to Rhône@Bognes. This results in a mean annual discharge of 30 m$^3$.s$^{-1}$,
contributing on average to 8 % of the Rhône@Bognes discharge with a pluvio-nival regime (the highest discharges occurring
in March with about 45 m$^3$.s$^{-1}$).

To reduce the influence of the marked glacio-nival or nival regime in the analysis of the discharges, we used the deseasonalised
anomalies of the mean daily discharges. Deseasonalised anomalies are computed for each day by comparing the targeted value
to the mean value on all of the corresponding days in the 1923-2010 period. For example, to obtain the deseasonalised
anomalies for the January 1st, we subtract the average of the 88 January 1st to each January 1st of the period 1923-2010. The
discharge data series from the three gauge stations are used on the 1923-2010 period because this is the common period
between gauge station series and ERA-20C reanalysis series (Table 1).
**2.2 The high-magnitude flood events**
The high-magnitude flood events are selected based on the percentile 99.9 value on the daily mean discharge of
Rhône@Bognes (1923-2010). The use of daily discharge series is consistent with the response time (1 day) of the upper Rhône
River catchment. Therefore, only days with a discharge greater than 1089 m$^3$.s$^{-1}$ are kept for the study. This results in the
identification of 38 days that correspond to 28 flood events since 6 flood events are characterised by consecutive days with
discharges upper than 1089 m$^3$.s$^{-1}$. For the flood events with consecutive days, the day with the peak discharge has been kept
to represent the date of the corresponding flood event. This set of 28 floods events (that correspond to at least 3-year return
period events) is considered to obtain a significant sample of flood events for the flood typology. Processes leading to the 5
largest events (greater than 3-year return period) will be separately treated in the discussion section. The first identified event
occurred on 19 August 1927 and the last happened on 14 January 2004. Among these 28 identified extreme flood events, the
floods of February 1990 and November 1944 were the largest events with daily mean discharge of 1 550 m$^3$.s$^{-1}$ and 1 480 m$^3$.s$^{-1}$
$^{1}$, respectively. The 1990 flood caused numerous damages in the upper Rhône River catchment, such as the destruction of two
bridges in the department of Haute-Savoie (France), of many roads and houses (DREAL report, 2011).





Lake Geneva may buffer flood discharges coming from the Geneva catchment because of its large size and its regulation. The
influence of this upper catchment on the floods recorded at Rhône@Bognes has then been tested by comparing percentile
values of discharges at Rhône@HDI, Arve@BDM and Valserine catchment when Rhône@Bognes discharges exceed the
percentile 99.9, i.e. for the 28 studied flood events. Discharges exceed the percentile 99.9 only once at Rhône@HDI, while
discharges exceed this percentile in more than 15 flood cases at Arve@BDM and Valserine catchment. This suggests that the
flows coming from the Arve and Valserine catchments play a dominant role for flood generation. Consequently, we mainly
focus on precipitation falling in the Arve and Valserine catchments to characterize the hydrometeorological processes that
triggered the 28 flood events at Rhône@Bognes.
**2.3 Precipitation data**
The daily precipitation series of the ERA-20C reanalysis (1900-2010; Poli et al., 2016) are used since it is one of the only
datasets covering the entire $20^{th}$ century. Daily precipitation accumulations falling in both the Arve and Valserine catchments
(about 2 900 km², called "A+V catchment precipitation" hereafter) have been estimated using i) the ERA-20C daily
precipitation at grid-points present in and around these catchments and ii) the Thiessen polygons method (Brassel and Reif,
1979). The Arve and the Valserine catchments are considered together because of the relatively low resolution ($1.125°$) of the
ERA-20C reanalysis compared to the catchment sizes.

An evaluation of the A+V catchment precipitation has been conducted through a comparison with an independent A+V
catchment precipitation computed from 17 meteorological stations (1950-2010) located in and around the Arve and the
Valserine catchments. The comparison revealed that the A+V catchment precipitation based on ERA-20C tends to be
underestimated, especially for the highest values (see Fig. A1 in the Appendix). However, daily precipitation percentiles from
the two datasets are in a good agreement (see Fig. A2 in the Appendix); when a high percentile value of precipitation
accumulation is observed for a given day in one of the datasets, a high percentile value is also observed for this day in the other
dataset. This suggests that the catchment precipitation distribution is correctly reproduced from the ERA-20C dataset, as
suggested by Rustemeier et al. (2019) at the monthly time scale in the Alps.  Therefore, only the percentile values of the A+V
catchment precipitation based on ERA-20C will be used.
**3 Indices for the flood typology**
**3.1 Indices from the precipitation sequences**
Different sequences of precipitation occurring prior to the floods have been tested to cover different types of floods such as
short-rain or long-rain floods (e.g. Merz and Blöschl, 2003). Two variables of the precipitation sequences have been
considered: (i) the sequence duration (number of days) and (ii) the ending day of the sequence. Fig. 2 illustrates the different
precipitation sequences tested for each of the 28 flood events: from 1 to 10 consecutive days (sequence duration) and for
sequences ending between 0 to 10 days prior to the flood day (temporality of the sequences). Then, the precipitation
accumulation of all sequences has been calculated and their respective percentiles estimated by sequence type to identify which
sequences better explained the 28 floods events (Fig. 3).

Higher percentile values are found for precipitation sequences ending one day before the flood events (D-1), whatever their
respective durations (Fig. 3). This result is in agreement with the response time of the Arve and the Valserine Rivers (i.e.





about 1 day) and with results of Froidevaux et al. (2015) for the Swiss macro-catchments (1 500 - 12 000 km²). Therefore,
sequences that end one day before the flood events seem to be the most relevant to explain the link between precipitation
accumulations and flooding.

The second step of the precipitation analysis aims to identify the sequence durations (between 1 and 10 days) that explain the
best the 28 flood events. The percentile values associated with each of these precipitation sequences are computed to identify
what sequence duration shows the highest percentiles. Fig. 4 shows the distribution of percentile values for sequences of
different durations (all ending the day prior to the flood date). A high dispersion and high mean values are observed for
precipitation sequences of 3 to 6-day duration, suggesting that these sequences are not relevant to explain the flood events.
Conversely, precipitation sequence of shorter (1-2 days) and longer (7-10 days) durations show a lower dispersion and higher
mean percentile values, suggesting that both may play a role in the generation of flooding. The independence of the
distributions between sequences from 1 to 10 days has been tested using Student-T and Z tests (0.95 confidence level). The
tests revealed that the mean percentile values of sequences of 1-2 days and 7-10 days are significantly higher than the ones
of sequences of 3-6 days. Therefore, short (1-2 days) and long (7-10 days) precipitation sequences seem to be two independent
and relevant factors to explain the occurrence of high-magnitude floods, given their high percentiles. Consequently, the choice
is between the 1-day and the 2-day sequences to characterise events associated with short precipitation sequences, and
between the 7-day, 8-day, 9-day and the 10-day sequences to characterise events associated with long precipitation sequences.
The same tests have then been applied to 1-day and 2-day sequence distributions and revealed no significant difference (at
the 0.95 confidence level). In addition, the median value of these sequences is not significantly different (Fig. 4). Given these
sequence characteristics, the 2-day sequences are considered to represent the short-rain episodes associated to flooding. We
also assume that 2-day sequences may reflect a more robust precipitation signal than the 1-day sequences. Hence, the 2-day
precipitation sequences (from D-2 to D-1 prior to the flood day) will be used as an index for performing the flood typology.
This is line with results of Froidevaux et al. (2015) highlighting that precipitation accumulating 0 to 3 days before the flood
is the most relevant factor for floods in Switzerland. Regarding the distributions of the long sequences (7-10 days), the tests
do not show any significant differences between them. However, 8-day precipitation sequences show the highest mean
percentile value and display the weakest dispersion. Thus, the 8-day precipitation sequences (from D-8 to D-1 prior to the
flood day) are considered to characterize the long rainfall episodes that seem to explain the occurrence of high-magnitude
floods and, thereby, they will be used as a second index for performing flood typology.
**3.2 Index from the discharge variability**
High water levels are sometimes observed many days prior to the flood events and this sometimes happens over longer periods
than the 8 days covered by the index of long-rainfall episodes. Thereby, the variation coefficient (VC) will be used a third
index to take into account this long-term high water stage preceding the discharge rise to the flood peak. It is computed
following Eq. (1):
$$VC_{(from\ D-7\ to\ the\ flood\ day)} = \frac{\sigma}{\bar{x}}, \tag{1}$$
where $VC_{(from\ D-7\ to\ the\ flood\ day)}$ is the discharge variation coefficient at Rhône@Bognes from D-7 to the flood day, $\sigma$ the standard
deviation of the discharge between D-7 and the flood day and $\bar{x}$ the mean discharge between D-7 and the flood day. VC is
computed from D-7 to the flood day to consider the 1-day response time of the catchment to the 8-day precipitation sequences.





## 4 Clustering and resulting flood typology

### 4.1 Hierarchical clustering

The hierarchical ascendant classification algorithm (Jain and Dubes, 1988) is used to identify the main flood types using the three above-mentioned indices. This algorithm tends to group individuals according to a similarity criterion that will be expressed in the form of a matrix of distances (Euclidean distance metric here). It expresses the distance existing between each individual taken two by two (Bruynooghe, 1978). The objective of this method is to divide a population into different classes by minimizing intraclass inertia and maximizing interclass inertia. A number of four clusters is retained from the hierarchical clustering algorithm because this partition displays the greatest relative loss of intra-class inertia.

### 4.2 Hydrometeorological characteristics of the flood types

The flood type 1 groups 9 flood events, all characterised by i) very high and stable discharge anomalies (about +400 $m^3.s^{-1}$ on average) from D-7 to D-2 , ii) low-to-moderate (percentiles lower or equal to 82) precipitation accumulations and iii) a mean peak discharge anomaly about +870 $m^3.s^{-1}$ (the lowest of the four types; Fig. 5). The type 2 groups 8 flood events characterised by i) a regular and large increase of discharge (mean anomaly from +150 $m^3.s^{-1}$ at D-7 to +500 $m^3.s^{-1}$ at D-1), ii) a similar increase in precipitation accumulations with percentile values from 62 (D-8) to 99.5 (D-1) and iii) a mean peak discharge anomaly near to +900 $m^3.s^{-1}$. The type 3 groups 6 flood events characterised by i) positive discharge anomaly lower than 200 $m^3.s^{-1}$ until D-3 , ii) significantly high precipitation accumulation at D-2 and D-1 (mean percentiles of 90 and 96.5) and iii) a mean peak discharge anomaly about +1 000 $m^3.s^{-1}$ (the highest with type-4 floods). Lastly, the type 4 groups 5 flood events characterised by i) anomalous low discharge until D-2 (about -50 $m^3.s^{-1}$ in average), ii) very high precipitation accumulations from D-2 to D-1 (mean percentiles of 99.1 and 99.7) and iii) a fast and large increase of discharge during two days until the flood peaks that reach about +1 000 $m^3.s^{-1}$ on average.

To test the dependence of the resulting flood typology to the ERA-20C precipitation dataset and its relative uncertainties (see section 2.3.), the same methodology was applied using the precipitation accumulations at the A+V catchment computed from (i) the ERA-20C and (ii) the station datasets over the common period 1950-2010, i.e. to 16 of the 28 flood events. Results obtained from the two datasets show very similar clustering with 3 groups (see Fig. B1 in the Appendix). Type 1 is no longer observed in these two classifications because most of type 1 flood events occurred before the 1950s (see section 4.3.). Types 2, 3 and 4 are very similar to the 3 groups obtained with these shorter precipitations series, supporting again the use of ERA-20C catchment precipitation for performing the flood typology.

### 4.3 Temporal characteristics of the flood types

Flood events of types 2, 3 and 4 mainly occurred during autumn and winter seasons (Fig. 6b). They are distributed over the whole 1923-2010 period without any clear cluster that would reflect flood-rich period (Fig. 6a). Regarding the flood magnitude, flood types 3 and 4 include the largest flooding with September 1927 (1 380 $m^3.s^{-1}$) and November 1944 (1 480 $m^3.s^{-1}$) for the type 3 and February 1990 (1 550 $m^3.s^{-1}$) for the type 4. Beyond these largest events, no clear change in flood magnitude can be observed since floods with the highest magnitude are observed at both the beginning and the end of the period. Conversely, flood type 1 is characterized by a strong seasonality with events occurring only in summer and beginning of fall (Fig. 6b). In addition, only one of the 9 events occurred after the 1950s. To understand the absence of this flood type from the 1960s, a





homogeneity test of Pettitt (Pettitt, 1979) has been applied to the daily discharge series in summer and beginning of fall at
gauge stations of Rhône@Bognes, Rhône@HDI and Arve@BDM. A break is detected in August 1961 in Rhône@Bognes (at
the 0.95 confidence level) with a decrease of 17% of the mean daily discharge (from 453 $m^3.s^{-1}$ to 378 $m^3.s^{-1}$) after the break.
No break is found in Arve@BDM, while a break is also detected in September 1956 in Rhône@HDI (at the 0.95 confidence
level) with a similar decrease of 16% of the mean daily discharge (from 335 $m^3.s^{-1}$ to 282 $m^3.s^{-1}$). Finding a quasi-synchronous
break in both Rhône@Bognes and Rhône@HDI gauge stations in a similar range of discharge suggests that the absence of
flood type 1 from the 1960s may result from this decrease in mean daily discharge upstream Lake Geneva. Since the change
abruptly occurred, the trigger is more likely related to changes in river management than climate. To identify if this change is
associated with the management of either Lake Geneva or the upper part of the catchment (Valais), the Pettitt test was also
applied to the discharge series (summer and beginning of autumn) of the Rhône River at Porte du Scex (Rhône@PDS), located
just upstream the lake (about 75 % of the lake's water supply; Grandjean, 1990). A break is detected in September 1956 in
Rhône@PDS (at the 0.95 confidence level), with a decrease of 20 % of the mean discharge (from 295 $m^3.s^{-1}$ to 236 $m^3.s^{-1}$ ).
Thereby, the break at Rhône@Bognes seems to be strongly related to discharge changes in the Valais catchment where 7 dams
have been built on tributaries of the Rhône River in the 1950s (Hingray et al., 2014). These dams have been built to store
summer water (high discharge due to glacio-nival regime) and release it mainly in winter for hydroelectricity production when
natural discharges are low and energetic needs are high (e.g. heating). Therefore, flood type 1 questions the hypothesis
previously formulated (see section 3.1) on the negligible role of the Geneva catchment discharges in the generation of extreme
flooding at Rhône@Bognes. Indeed, the large Lake Geneva and its regulation buffer flood peaks but made flood generation at
Rhône@Bognes easier until the 1960s by providing summer high discharges. This contribution of the Geneva catchment will
be further discussed in the following section.

## 5 Discussion

Type 1 floods are associated with moderate precipitation that does not exceed mean percentile values of 82 and, thereby, that
cannot alone explain the flood occurrences (Fig. 5). A detailed analysis of the respective contribution of the three catchments
reveals that the Geneva catchment plays a dominant role by providing abnormally high discharges (about +400 $m^3.s^{-1}$; Fig.
7c), contributing to more than 50 % of the discharge at Rhône@Bognes (Fig. 7b). To understand the reason of these abnormally
high and long-lasting discharges, the Geneva catchment precipitation has been computed using the same method as for A+V
catchment precipitation (Fig. 7c). The resulting catchment precipitations appear very similar to the A+V catchment
precipitation and they cannot explain the high and long-lasting discharges either. In addition, for 5 of the 9 type 1 events,
abnormally high discharges coming from the Geneva catchment lasted 40 to 120 days. Such high water stages in late summer
may then result from particularly intense melting of the numerous and large glaciers of the Valais. Indeed, the Mer de Glace
and Argentière glaciers have been affected by important losses in mass balance during the summers the type 1 floods occurred
(Vincent et al., 2009, 2014). These glaciers are located a few tens of kilometres from the Valais but glacier losses are expected
to be regionally similar (Huss, 2012). These observations support a glacial trigger of the abnormally high, long-lasting summer
discharges that make flooding possible without heavy precipitations. Soil saturation also seems to influence the generation of
flood type 1. Indeed, a precipitation episode occurred a few days before the flood event (around D-6) triggering only a discreet
increase of discharge, while a second episode (D-1) with similar precipitation accumulation led to the flood peak (Fig. 7c).
This suggests that soil infiltration has buffered the first precipitation episode, while soils were saturated for the second one,
promoting runoff and, thereby, a stronger hydrological response. On another hand, the increase of discharge is due to the
contribution of the Arve River and in a larger part to the contribution of the Valserine River that flows from the Jura massif,
an area where soil saturation has been recognized as a key process for flood generation (Froidevaux et al., 2015). The role of



soil saturation has not been clearly observed for the other flood types. This may be related to the seasonality of the other flood
types that occurred in late autumn and winter, i.e. during the rainiest and colder season that makes soils often saturated and
that limits soil evaporation. Summer and beginning of autumn (season of type 1 floods) are rather dry periods and the soils are
more sensitive to moisture variations. Therefore, type 1 floods seem to result from a combination of i) intense ice-melting that
triggers a high, long-lasting discharge baseline, ii) a moderate precipitation episode lasting a few days (D-8 to D-5) saturating
soils and iii) a moderate-to-high precipitation episode the day preceding the flood peak. Compared to other flood types, the
secondary role of precipitation in flood type 1 generation is well highlighted in Fig. 8. This flood type 1 is new compared to
previous flood typologies (e.g. Merz and Blöschl, 2003; Sikorska et al., 2015; Brunner et al., 2017; Keller et al., 2018). Flood
type 1 may look similar to the "glacier-melt floods" type of Sikorska et al. (2015) and Brunner et al. (2017) because of the
glacial component. However, this does not include the moderate precipitation episode needed to trigger type 1 flood event.

The autumn-winter type 2 flood events are associated to (i) moderate but increasingly large precipitation accumulations (mean
percentiles increasing from 70 to 90) between D-8 and D-3, triggering a regular increase of discharge and to (ii) heavy
precipitation accumulations (mean percentiles from 90 to 99,5) from D-2 to D-1, triggering the flood peak (Figs. 7a and 7c).
The high discharges in Rhône@Bognes from D-7 to D-1 (from +150 $m^3.s^{-1}$ at D-7 to +500 $m^3.s^{-1}$ at D-2) are mainly provided
by the Geneva and Arve catchments (43 % and 36 %, respectively; Fig. 7b). The contribution of the Arve catchment from D-
1 to the flood day explained 45 % of the floods events (Fig. 7b). Therefore, a combination of both long and short rain episodes
seem to well explain the generation of flood type 2 as confirmed by the position of all type 2 floods in upper right-hand corner
of Fig. 8 (i.e. high percentiles of both short and long percentile sequences). Flood type 2 results from the combination of short
and long intense precipitation sequences. Thus, this is very similar to the "long-rain floods" type defined by Merz and Blöschl
(2003), Sikorska et al. (2015) and Brunner et al. (2017) as events triggered by i) rainfall over several days that saturates the
catchment and cause high discharge conditions and ii) additional heavy rainfall that generates the flood peak. Compared to
Keller et al. (2018), flood type 2 is closed to their "long duration floods" characterized by high precipitation depths and
embedded episodes of high precipitation intensities.

Conversely to flood types 1 and 2, discharges anomalies of autumn-winter flood type 3 are low from D-7 to D-2 (below
+200 $m^3.s^{-1}$ on average, Fig. 7c). Type 3 flood events are mainly triggered by precipitations the two days before the flood
(mean percentile values from 90 to 96,5; Fig. 7c). The Arve catchment contributes to about 45 % of the flood peaks (Fig. 7b).
Compared to types 2, flood peaks of type 3 are larger on average, while precipitations accumulated the two days preceding the
flood reach higher percentiles for type 2. In addition, percentiles of both short and long precipitation episodes show lower
values than those triggering flood types 2 and 4 (Fig. 8). This suggests that precipitation alone cannot fully explain type 3 flood
generation. Ice melting is unlikely at this season and soils are expected to be wet to saturate since this season is rather wet and
cold. Snowmelt is the most probable candidate since a large part of the catchment may be covered by snow and sensitive to
changes in temperature. Therefore, flood type 3 seems to result mainly from short intense precipitation sequence as well as
probably snowmelting. Snowmelting, however, acts a minor role compared to ice melting in flood type 1 generation (Fig. 8).
The precipitation characteristics makes this type 3 similar to the "short-rain floods" (Merz and Blöschl, 2003; Sikorska et al.,
2015; Brunner et al., 2017) or to the flood type "shorter duration events with higher precipitation intensity" (Keller et al., 2018)
that results from rainfall of short duration but high intensity. These types, however, do not include the snow component. Type
1 could thus be an intermediate case between the "short-rain floods" and the "rain-on-snow floods" of Merz and Blöschl
346 (2003).






For flood type 4, percentiles of long precipitation episode are low (mean percentile values lower than 50, Fig. 7a) and
abnormally high discharges have not been identified from the D-8 to D-3 period (Figs. 7b and 7c). By contrast, really high
precipitation accumulations fell from D-2 to D-1 (mean percentile value upper than 99; Figs. 7a and 7c), leading to flooding
in Rhône@Bognes. The contribution of the Geneva catchment is very low (< 10 %), while the ones of the Arve and the
Valserine catchments reach respectively values higher than 40 % (Fig. 7b). For example, during the flood of February 1990,
the peak discharge of the Valserine River reached 360 $m^3.s^{-1}$ (value estimated higher than a 50-year return period discharge;
MEEDDAT DREAL RHONE-ALPES, 2011), while the Geneva catchment played a weak role due to the regulation and/or a
slower response to these heavy precipitations. Therefore, flood type 4 result from heavy short precipitation episode (Fig. 8),
corresponding well to the "short-rain floods" type with a duration of heavy precipitation (2 days) longer than in the definition
(e.g. Merz and Blöschl, 2003).

Lake Geneva catchment was first assumed to plays a negligible role of flood generation at Rhône@Bognes because of the
numerous hydraulic infrastructures in the Valais, the large size of the lake and its regulation that buffer the discharge variability.
Nevertheless, the Geneva catchment may contribute to the flood generation by providing high water level downstream over
longer time scale than the typical one of flood generation as identified by Froidevaux et al., (2015). This contributes
significantly to type 1, in a lesser extent to types 2 and 3 and not to type 4 flood generation.

This flood typology aims first to explore in what extent the generation of high-magnitude flood events can be explained by
precipitation only. However, performing the flood typology required taking into account a sufficiently large sample of flood
events that encompasses relatively frequent flooding (from 3 year return period). Based on the process knowledge gained from
the flood typology, flood process generation of events with the highest magnitude can be discussed (Fig. 8). The largest flood
event (February 1990, 100-year return period; Evin et al., 2019) is associated to type 4 and it is characterized by the heaviest
precipitation with the most extreme percentile of short precipitation sequence (99.99). The 20-year return period event (number
1-4 on Fig. 8) are of types 2, 3 and 4, suggesting mixed processes possibly including snowmelt. A closer look to the
precipitation features, however, reveals very high percentile of the short precipitation sequences (>99.2) for various percentiles
of long precipitation sequences (between 78.8 and 99.5). Precipitation accumulating the two days before an event seem thus
to be the most relevant for the highest-magnitude (>20 return period) events. Regarding 10-year return period events, they are
of types 1, 3 and 4. These events are scattered in Fig. 8, with most of them characterized by very high to extreme percentiles
(>99.5) of short precipitation sequences (Fig. 8). Events with the 5[th], 7[th] and 8[th] highest magnitude are the exceptions and
belong to types 1 and 3. Therefore, 10-year return period events cannot be systematically attributed to heavy precipitation
sequences and other processes such as ice or snow melt should be taken into account. This result slightly differs from
observations of Merz and Blöschl (2003) suggesting a dominant role of precipitation for generation of >10-year return period
events. This difference might be partly explained by i) the presence of numerous and large glaciers in our catchment that may
play an important hydrological role and ii) the large size of our catchment. Finally, the two precipitation indices appear to be
less and less relevant when considering all flood events, even more when considering annual maximum discharges. This
suggests that the variety of processes involved is higher when considering low-to-medium magnitude events or, inversely, that
higher is the magnitude considered, higher the role of precipitation accumulation is.





## 6 Conclusions

A typology of >3 year return period flood events occurring between 1923 and 2010 in the large and mountainous catchment of the upper Rhône River has been performed through three indices based on precipitation (2-day and 8-day precipitation accumulations) and discharge (variation coefficient) series. This resulted in four types:

i)   type 1 floods resulting from a combination of i) intense ice-melting that triggers a high, long-lasting discharge baseline, ii) a moderate precipitation episode lasting a few days (D-8 to D-5) saturating soils and iii) a moderate-to-high precipitation episode the day preceding the flood peak.

ii)  type 2 results from the combination of short and long intense precipitation sequences, similar to the "long-rain floods" type defined by Merz and Blöschl (2003),

iii) type 3 seems to result mainly from short, intense precipitation sequences as well as probably snow melting,

iv)  type 4 result from heavy short precipitation episode (Fig. 8), corresponding well to the "short-rain floods" type with a duration of heavy precipitation (2 days) longer than in the definition (e.g. Merz and Blöschl, 2003).

Therefore, 2 types are directly related to precipitation accumulation, while the two other types cannot be explained by precipitation only and involved also other processes such as ice or snow melt. The typology also revealed that Lake Geneva and its catchment can play a key role on flood generation by providing a discharge baseline. This was particularly the case during certain summers of intense ice melting. Additional moderate rainfalls have thus led to high-magnitude flood events (type 1). However, this flood type has not been longer observed since the building of dams in the 1950s for flow regulation and hydroelectric production.

The four events of highest magnitude (>20 year return period) are of various types but are all triggered by heavy precipitation during the days preceding the floods. Considering floods of weaker magnitude progressively shows a decreasing role of the precipitation accumulations, suggesting a higher diversity of involved processes in the generation of e.g. annual flooding.

Since two flood types and/or the high-magnitude events are directly explained by atmospheric variables (i.e. multi-day precipitation sequences), our results open new perspectives for flood hazard assessments directly based on climate model outputs. This, however, requires first identifying robust atmospheric predictors of heavy rain accumulation. Finally, the successful evaluation and use of the ERA-20C meteorological reanalyses to assess precipitation accumulations over the last century encourage transposing such studies in any other region, where long discharge series are also available.




**Appendix A: Comparison between the daily cumulative precipitation provided from the ERA-20C reanalysis and**
**from the stations**
The evaluation of the precipitation at the A+V catchment scale from the reanalysis ERA-20C over the 1950-2010 period (in
comparison with the precipitation from the 17 meteorological stations from the METEOFRANCE and the METEOSWISS
organisms, located in and around the catchments of Arve and Valserine) is shown in Fig. A1. The ERA-20C precipitation at
the A+V catchment scale tends to underestimates the daily cumulative precipitation value, in comparison with the gauge
precipitation, especially for the highest values.

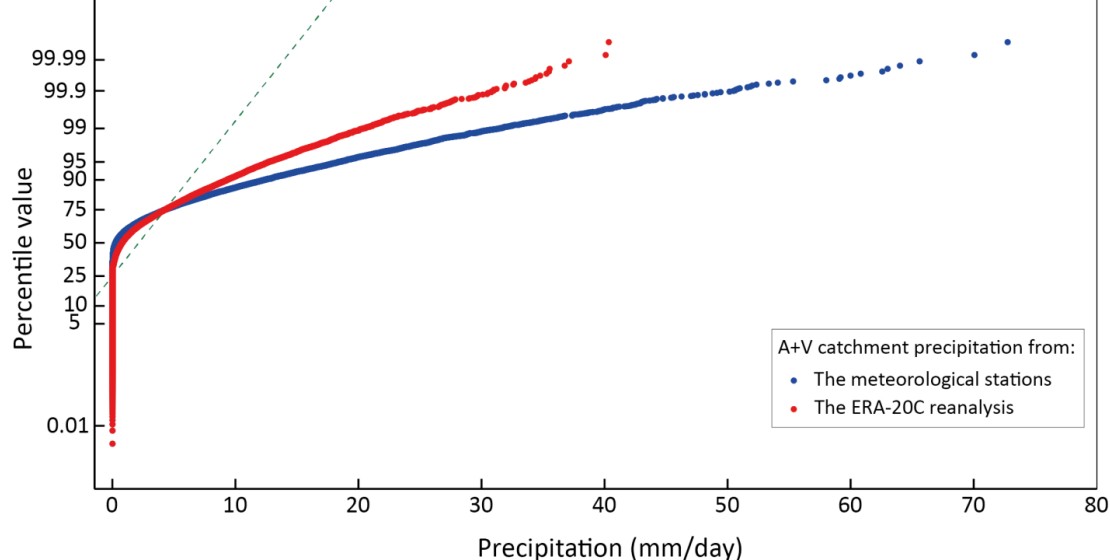


**Figure A1**. Normal probability diagram comparing the distribution of the precipitation at the A+V catchment scale from the reanalysis ERA-
20C (in red) and from the meteorological stations (in blue).



While ERA-20C tends to underestimate the daily cumulative amount at the A+V catchment scale, the two distributions of
the precipitation intensity are in good agreement (Fig. A2).



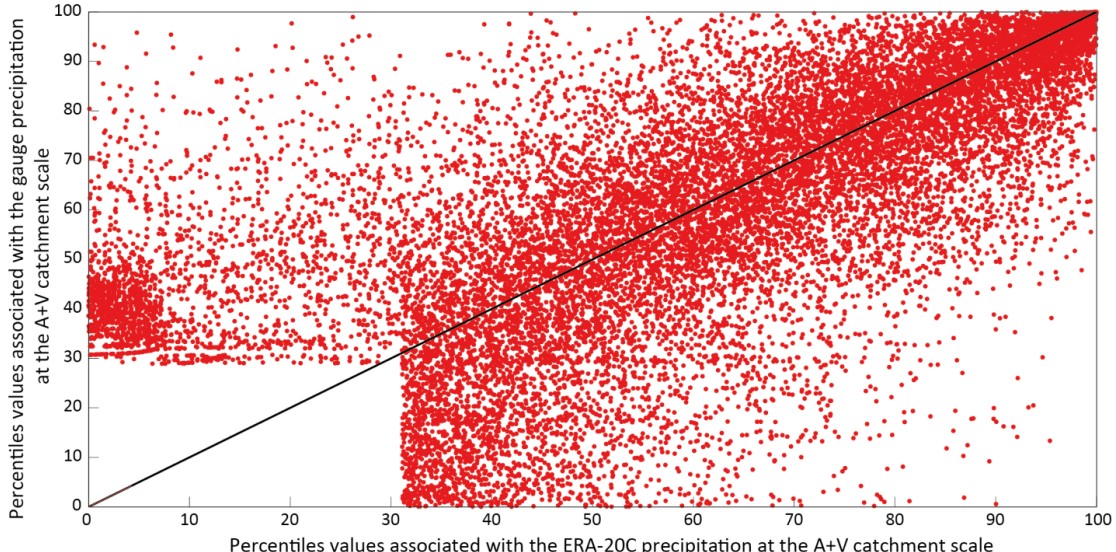



**Figure A2**. Percentile/percentile diagram of the daily precipitation at the A+V catchment scale from the ERA-20C reanalysis and from the meteorological stations, over the 1950-2010 period. For each day, the percentile value associated with the daily precipitation amount is given, in regard to the respective distribution of the precipitation from the ERA-20C reanalysis and from the meteorological stations. The percentile values before 29 (stations) and 31 (ERA-20C) are associated with dry days.








**Appendix B: Comparison of the flood types built by the use of the ERA-20C and the raingauges at the A+V catchment scale**

The last step of the evaluation of the ERA-20C precipitation at the A+V catchment scale consist to compared the results of the
two flood type classifications, based on the two data-sets within the common period 1950-2010. The results are shown in Fig.
B1. The hydrographs resulting from these two classifications present similar trends: the 3 flood classes detected here, called
classes 2, 3 and 4 (in reference to the classes detected during the 1923-2010 period), are very similar for both classifications.
Previous flood class 1 is no longer observed in these two classifications because as said in the article, 7 out of 9 flood events
of the class 1 occurred before 1950.



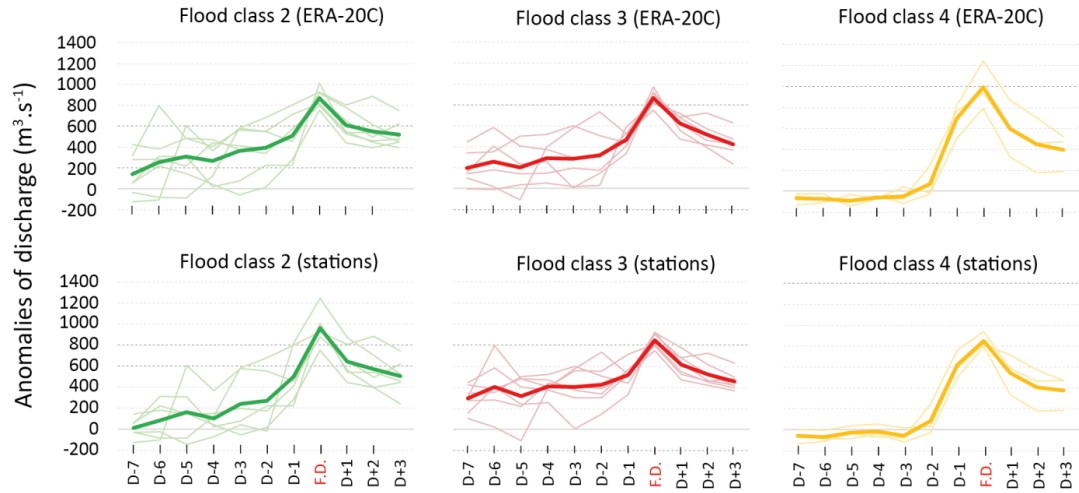

**Figure B1**. Comparison of the flood types from the classification based on the precipitation index from the A+V catchment precipitation from the ERA-20C (upper panel) and from the meteorological stations (lower panel).

*Author contributions*. All of the authors helped to conceive and design the analysis. FR performed the analysis and wrote the manuscript. BW and SA participated to the analysis, commented on the manuscript and contributed to the writing of the paper.

*Acknowledgements.* This work is a contribution to the Cross Disciplinary Program "Trajectories", from the Grenoble University. Within the CDP-Trajectories framework, this work is supported by the French National Research Agency in the framework of the "Investissements d'avenir" program (ANR-15-IDEX-02). The authors are grateful to the Compagnie Nationale du Rhône (CNR) and to the Federal Office for the Environment of Switzerland (FOES), for providing discharge measurement data series from gauge stations located in France and Switzerland. The authors are also grateful to the MeteoSwiss and to the MeteoFrance organisations, for providing observation precipitation data series from meteorological stations located in France and Switzerland.

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





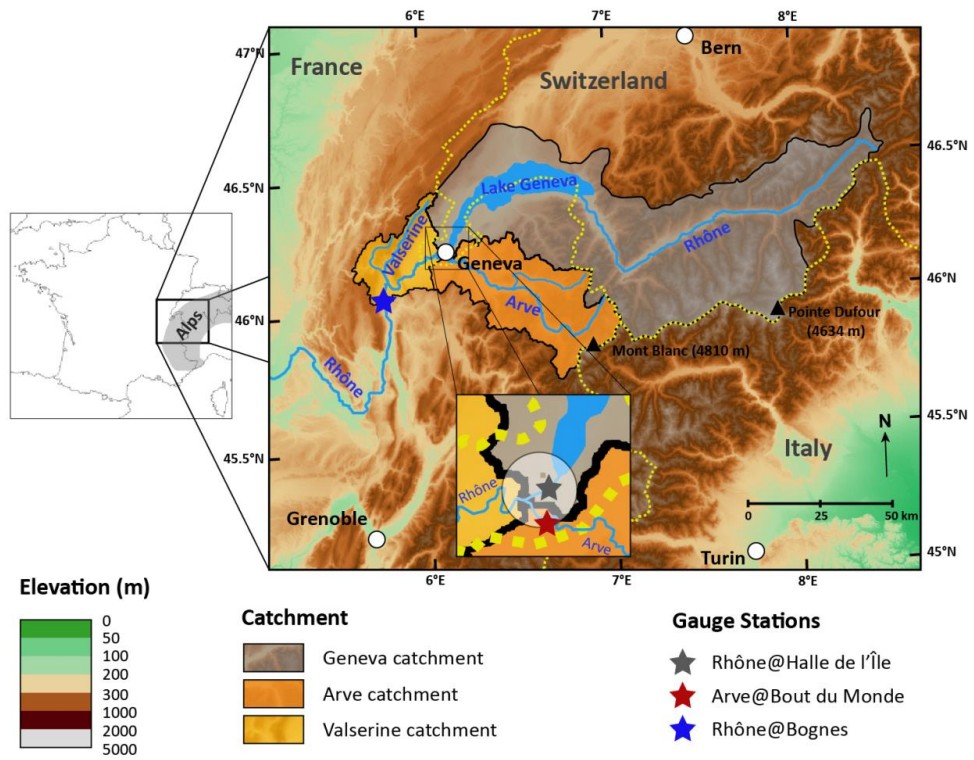


**Figure 1: Location map of the studied upper Rhône catchment located in the French and Swiss Alps. The map also shows the division**
**in three sub-catchments and the three gauge stations used in this study.**





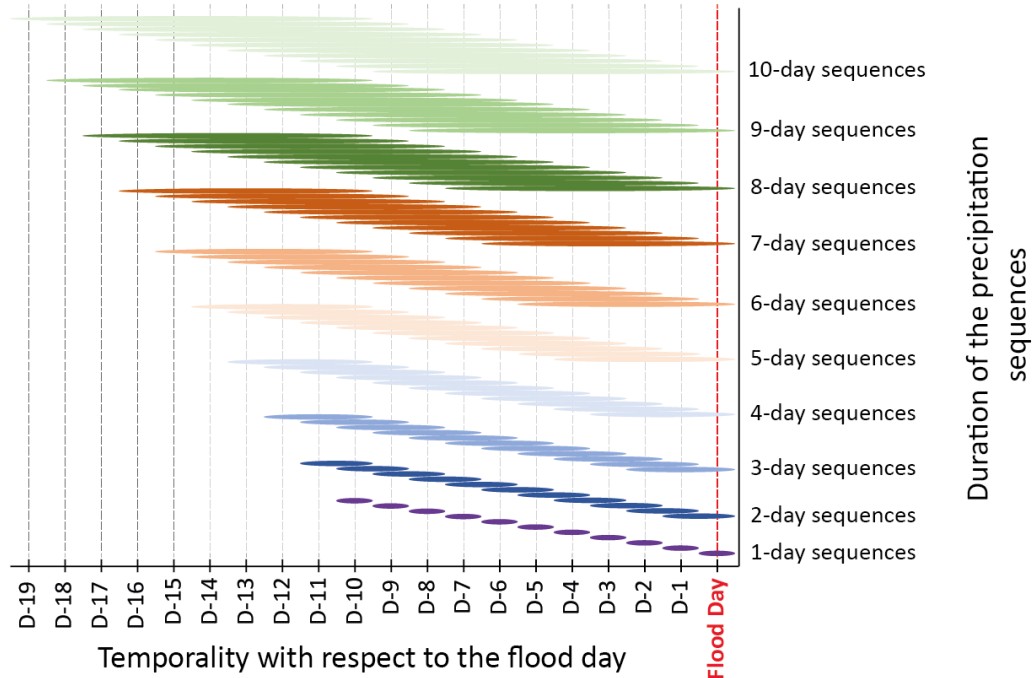


**Figure 2: Conceptual graphic of the different precipitation sequences analysed for each of the 28 flood events. The two sequence**

**parameters are (i) the sequence duration, from 1 day to 10 consecutive days (y axis) and (ii) the temporality of the sequences, i.e.**

**the ending date of the sequences, from 10 days prior to the flood day, to the flood day (x axis). The colour code is used to distinguish**

**sequences of different durations.**

598



599

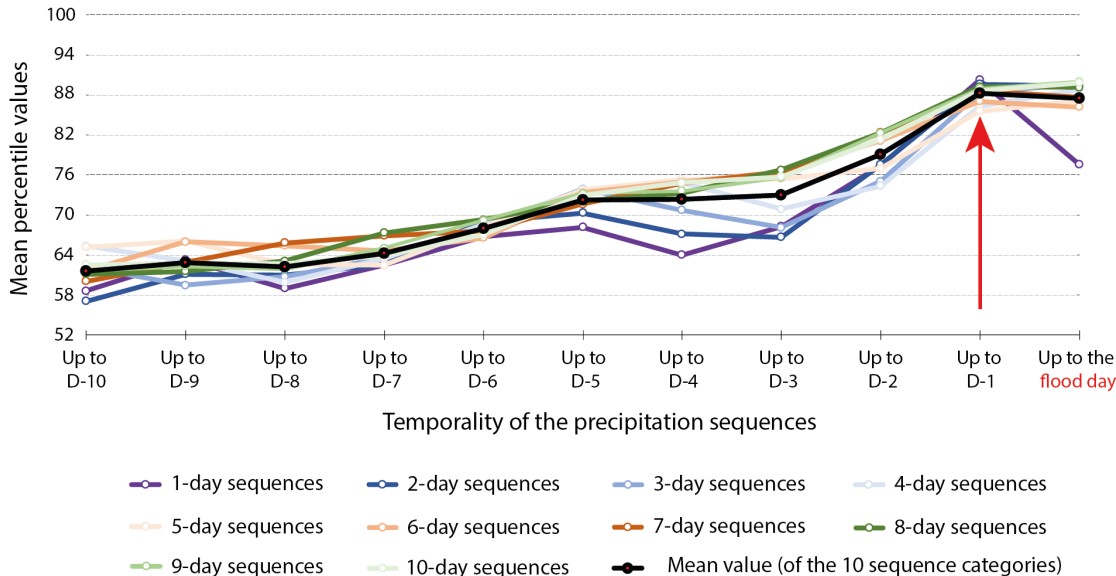

600

**Figure 3: Mean percentile values of the precipitation accumulations calculated for precipitation sequences of various durations (1
to 10 days) and various temporalities (ending between 10 and 0 day prior to the flood day). Each colour line corresponds to the
mean percentile of sequences of a given duration obtained for the 28 flood events. The colour code used to distinguish the different
sequence durations is similar to Fig. 2. Each point corresponds to the ending date (temporality) for which the mean percentile of
the sequences of a given duration has been computed. The mean percentile value of all sequences together is shown in black.**




















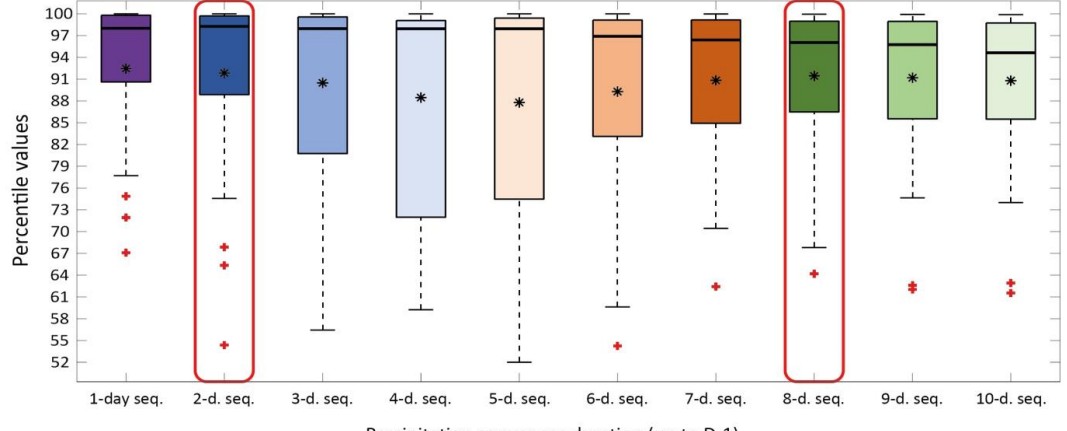


**Figure 4: Boxplot of percentile values for precipitation sequences of various durations (colour code identical to Fig. 2 and Fig. 3), all ending the day prior to the 28 flood dates. Bottom and top of box plots correspond to the first and third quartiles, respectively. The black band inside the box corresponds to the median and the black star to the mean. Red crosses below the box plot indicate extreme values (out of the whisker) and the ends of the whiskers represent the lowest datum still within the 1.5 inter-quartile range of the lower quartile, and the highest datum still within 1.5 inter-quartile range of the upper quartile.**






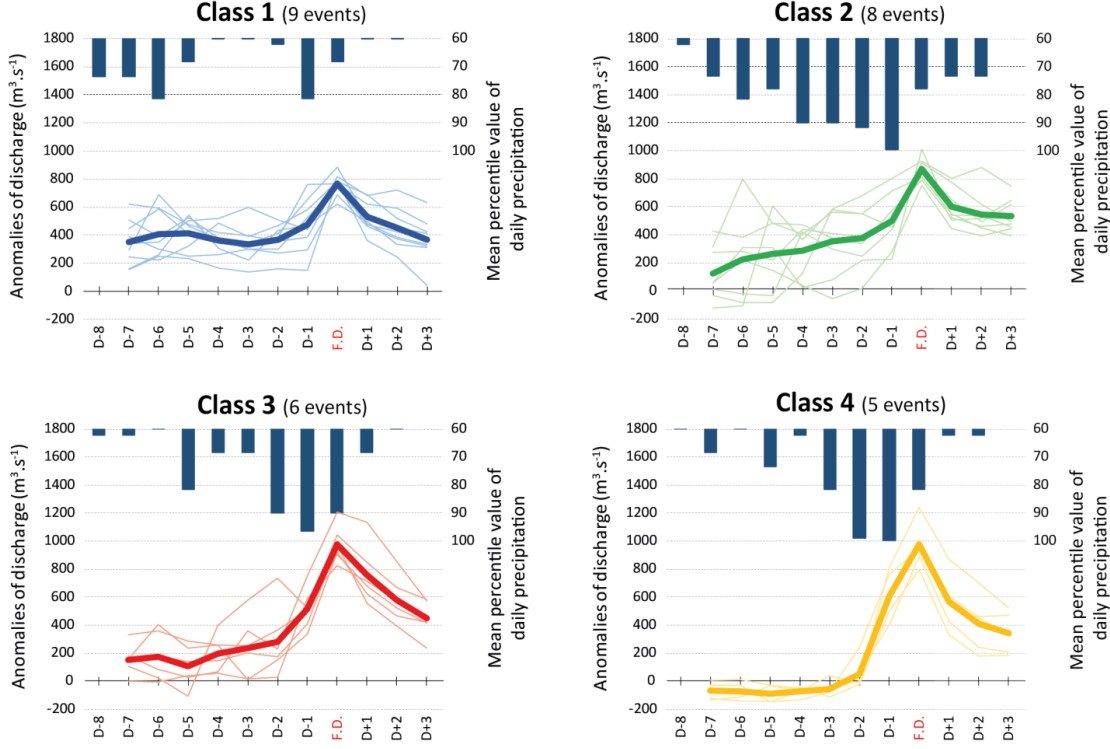

**Figure 5: Hyetograms and hydrographs associated to each of the four flood types. Hydrographs are expressed as deseasonalised**
**anomalies of discharge for each of the flood events (thin curves) and on average (thick curves). Hyetograms show the mean percentile**
**values of daily precipitation.**




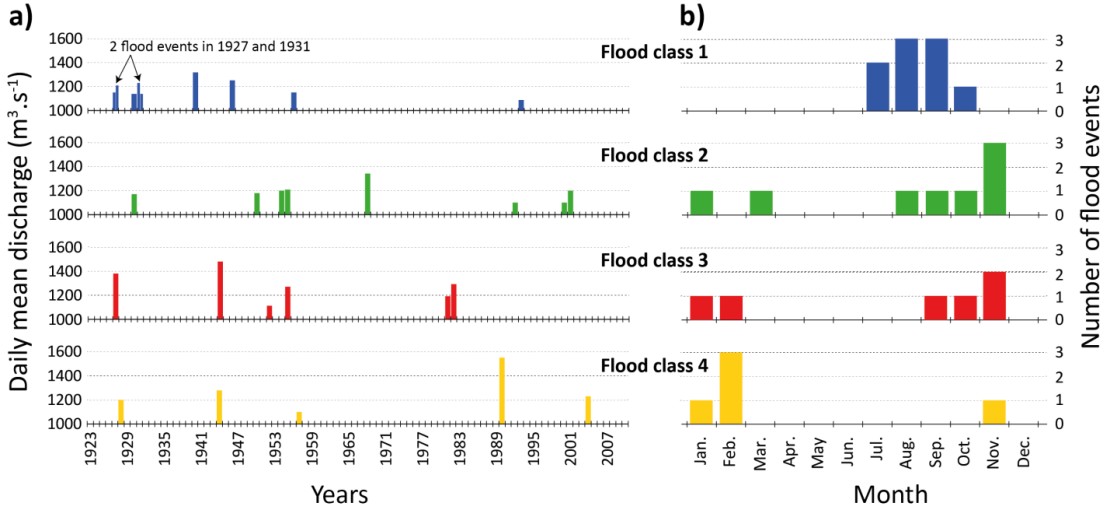

**Figure 6: (a) Chronology (with daily discharge) and (b) seasonality of the 28 flood events grouped by flood type.**




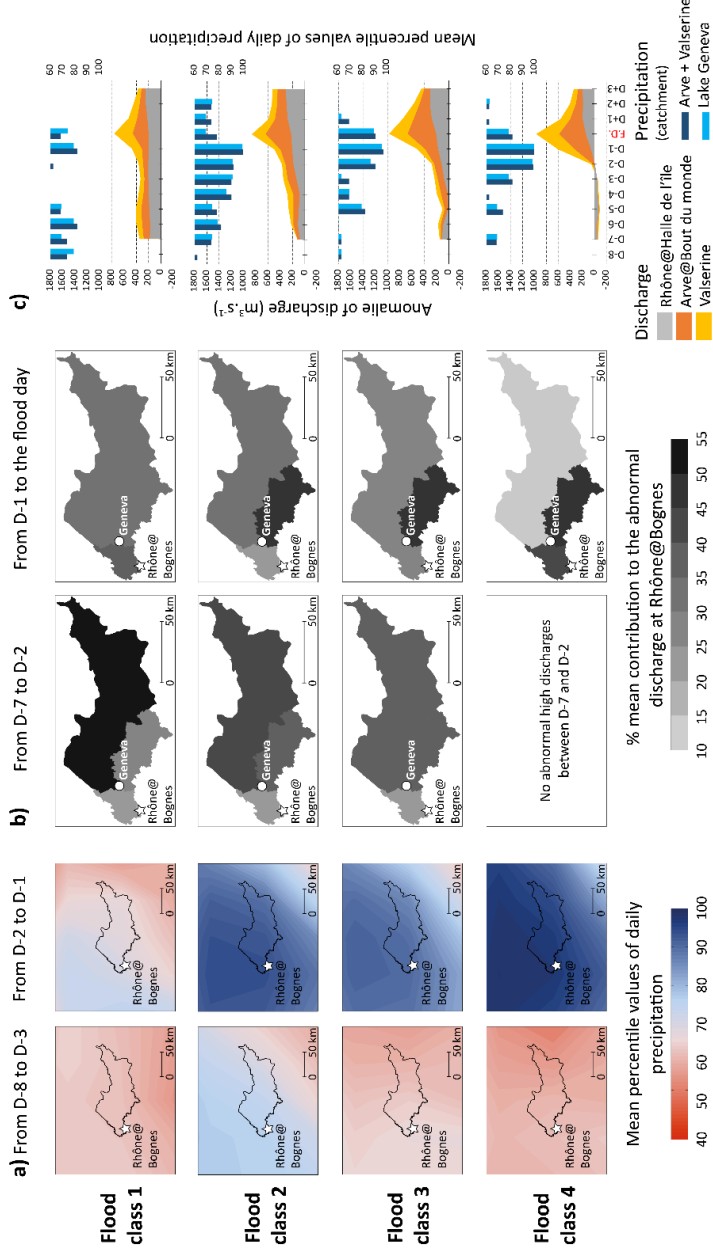



**Figure 7: For each of the four flood types: a) map of the mean percentile values of daily precipitations; b) contributions for each of**
**the sub-catchments to the abnormally high discharges observed in Rhône@Bognes; c) hydrographs and hyetograms associated to**
**the flood types. The mean contribution of the sub-catchment is given for the global periods D-7 to D-2 and D-1 to the flood day. The**
**hyetograms show the mean daily percentile values of daily precipitation for the A+V and Geneva catchment precipitation. The**
**hydrographs show the daily mean deseasonalised anomaly of discharge for each sub-catchment. The accumulation of the three**
**abnormally give the daily mean deseasonalised abnormally observed in Rhône@Bognes.**



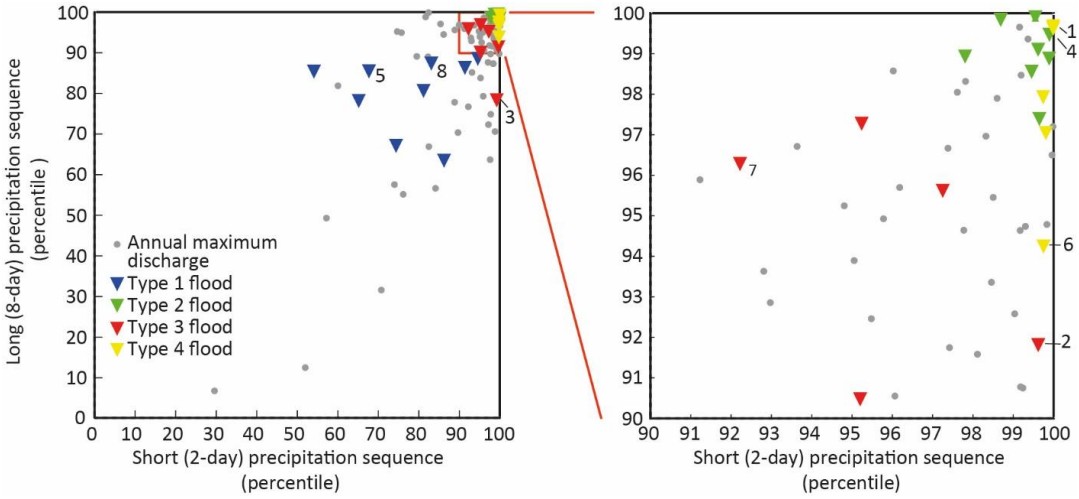


**Figure 8: Percentiles of short (2-day) versus long (8-day) precipitation sequences for each type of floods, as well as for annual**


**maximum discharge. Numbers show the eight largest floods of the last 88 years, i.e. greater than 10 year return period events. The**


**right-hand panel is a zoom of the diagram for percentile values higher than 90.**


























**Table 1: Name, number, organization name, river and starting year of the three gauge stations used in this study.**

| Station name | Number | Organization name | River concerned | Starting year |
|---|---|---|---|---|
| Rhône@HDI | 2606 | Federal Office for the Environment (FOEN) | Rhône | 1923 |
| Arve@BDM | 2170 | Federal Office for the Environment (FOEN) | Arve | 1904 |
| Rhône@Bognes | V1020010 | Compagnie Nationale du Rhône (CNR) | Rhône | 1920 |
