# Peer review of "The role of precipitation for high-magnitude flood generation in a"

_Hydrology and Earth System Sciences, 2019_

## Referee Comment (RC1) · Anonymous Referee #1 · 14 Mar 2019

Review of "The role of precipitation for high-magnitude flood generation in a large mountainous catchment (upper Rhône River, NW European Alps)"

This paper describes the processes underlying floods that exceed the 3 year return period for the Rhône River over the period 1923-2010. This is done based on the following steps: i) identify floods events with a return period greater or equal to 3 years, ii) compare to which rainfall durations and timing best explain these floods, iii) provide flood topologies based on these comparisons and some cluster analyses. It is concluded that there are four flood types, which are described.

I have several major concerns that at present inhibit me from recommending publication

of this work in HESS:

*) It is unclear what we really learn. For example, the finding, which report correlations between floods and precipitation characteristics, these findings are only related based on the correlation of some floods of some catchment with some arbitrarily chosen P characteristics. This is potentially interesting, but does not answer any clear question to me, nor does it support the claim that "our results open new perspectives for flood hazard assessments directly based on climate model outputs.". Sure rainfall is important in driving this floods, but this seems trivial, and I would expect some clear insight beyond this trivial statement. The provided topology of floods is an interesting way to summarize the type of flood events, but I do not see how it is really relevant beyond the case provided here.

*) I am unconvinced the ERA data is accurate enough to do any kind of the inferences that are performed in this study. While it is argued that the data is accurate enough (to use percentiles), as supported by Fig A2, this figure to me shows a lot of scattering which lead to the conclusion that these data are in fact highly unreliable to use.

*) Why is soil moisture not considered in explaining the floods? Could this be ruled out a-priori for some reason?

*) The writing of the paper is often unclear or inaccurate. I made some suggestion below, but this list is far from exhaustive.

*) I found it hard to follow the figures, which I made some comments on below.

Detailed comments

L11: Drop "High-impact climate events such as" because you can simply state "Floods are highly destructive natural hazards causing widespread impacts on . . ." which makes it clear you focus on floods in this paper.

Line 12: "which limits reliable predictions", not "which limiting reliable prediction."

L13 "takes advantage of centennial-" or simply "uses".

L14: drop "the" before "high-magnitude".

L14: "the 99.9th percentile" not "percentile 99.9".

L17: "the amplitude of" seems redundant.

L29: "the frequency and magnitude" not "frequency and magnitude".

L34: drop the word "poor" otherwise the sentence is not logical. (Or change the word "limited" but that seems more complicated).

L36-37: Maybe it is my limitation, but I have no idea what a "socio-ecosystem" really refers too. Can you not just replace this by a less fancy sounding, but more straightforward word?

L40-41: "catchment characteristics, and air temperature, etc are not processes, as this sentence does imply.

L41-44: "Considered flood events . . . process-based flood types". This sentence needs to be rewritten

L45: rewrite to "lasting a maximum of one day".

L49: where does "2002" refer to?

L54: "per year" not "per years".

L39-63: This section seems to contain many irrelevant details on what is done (e.g. list of numbers of how many catchments are studied), rather than focus on "what is learned". This is not wrong, but in does not enhance the readability of the paper.

L69-70: what are "lesser informed events". Do you mean "event for which little hydrological data is available, and thus alternative sources of information need to be inquired"?

L70-71: how can both be "dominant" at the same time? Do you mean "important" instead? It is also unclear what "dominant" really means in this context.

L72-75: It is unclear what is being said here (and it seems to be a rather important statement).

L75-84: I do not see the relevance of this speculated reason. It is one of the many reasons that these floods have has less attention. I would just remove it.

L88: "to what extent" (not "in").

L88-90: this not only assumes that snow and ice are not important. It seems to assume that (antecedent) soil moisture is not important. . .

L97: "influence" is redundant.

L99: "Rainy days represent 30 to 45 %" is clear, but oddly phrased.

L99-100: What do you mean by "reaching 45 to 105 mm/day on average" what are these numbers based on?

L101:-102: are these mean monthly values? (I suspect that showing the seasonal hydrographs of the three catchments provides clearer and more concise insight into the differences in their discharge regimes.) This will also show if the discharge estimated in L130-135 looks reasonable.

L132: "it" not "its"

L137-142: It is unclear to me why this normalization has been performed. I.e. what do you really mean by "To reduce the influence of the marked glacio-nival or nival regime in the analysis of the discharges"? And why would you want to do that?

L148: "exceeding" not "upper than".

L173-181: You state that "[. . .], daily precipitation percentiles from the two datasets are in a good agreement (see Fig. A2 in the Appendix); when a high percentile value of

precipitation accumulation is observed for a given day in one of the datasets, a high percentile value is also observed for this day in the other dataset.". However, when I look at Figure A2 I find this difficult to conclude with so much scatter.

L186-189: I do not understand the description and meaning of Figs 2 and 3.

L198-199: "that best explain".

---

## Referee Comment (RC2) · Anonymous Referee #2 · 25 Mar 2019

The manuscript presents a method for linking precipitation data and flood discharge data to classify flood types of the upper river basin of the Rhone River. The main aim is to shed light on precipitation characteristics of high-magnitude floods. The study describes a complex case of a main river that is fed by a regulated lake and two tributaries. Moreover, the river basin is influenced by snow and glaciers, and more importantly by reservoirs for hydropower generation. The authors use the ERA-20C dataset for analysing the precipitation characteristics.

The study presents some interesting findings. On the one hand, the study shows an interesting approach to assess flood typologies (hierarchical clustering) in a river basin

with a relevant human influence by lake regulation and hydropower reservoir operation. On the other hand, it provides insights in flood generation processes.

However, some improvements are suggested before the publication of the manuscript.

- The main findings could be a bit more highlighted and generalized.

-It would really be interesting to look at the precipitation patterns leading to the superimposition of the flood peaks from the three subcatchments. The topology of the river network and the buffering effect of Lake Geneva suggests that the precipitation patterns (spatio-temporal distribtion of rainfall in the three subcatchments) leading to superimpostion of flood peaks and thus to high-magnitude floods must follow a certain pattern. I suggest to elaborate more on that in the discussion when introducing the "type 1 floods" (see, e.g., Pattison, I., Lane, S. N., Hardy, R. J., and Reaney, S. M.: The role of tributary relative timing and sequencing in controlling large floods, Water Resour. Res., 5444–5458, doi:10.1002/2013WR014067, 2014. or Zischg, A. P., Felder, G., Weingartner, R., Quinn, N., Coxon, G., Neal, J., Freer, J., and Bates, P.: Effects of variability in probable maximum precipitation patterns on flood losses, Hydrol. Earth Syst. Sci., 22, 2759–2773, doi:10.5194/hess-22-2759-2018, 2018).

Minor remarks:

-line 341: In my opinion, the statement on the role of snowmelting cannot be concluded from the present study. The effect of snowmelt was not analysed.

-line 371: As above, the role of snowmelt, although as mixed process, cannot be stated without having analysed it in detail.

-line 394: same as above.

-lines 408-410: Please describe what you mean with "new perspectives".

---

## Referee Comment (RC3) · Anonymous Referee #3 · 7 Apr 2019

The manuscript discusses how flood events can be categorized into a typology based on an understanding of the processes that dominated in a catchment. The authors considers a mountainous area in the Alps for their study. The authors uses a long series of daily rainfall records, re-analysis of this data, and flow records from three gauges in the catchment.

The manuscript is difficult to follow. The flow data are first described as coming from three catchments, then two, and finally one catchment, being runoff from a heavily regulated catchment where the considered flow is the combined flow from two of the smallest catchments. The data are described as being adjusted for seasonal variation,

but all results are reported as flow values. With respect to the precipitation two sources are available, station data and ERA-20. I have not worked with ERA-20, but I expect that it - like the other ERA reanalysis products - to some extend are based on measured station data for precipitation, at least for some of the years. Hence the study most likely uses and compares different data products on which on is based on the other. Please clarify what data sources are used in the study and how they are used. I dont understand how Figure 2 can be used to deduct exactly which data are used in the clusster analysis and how.

I also find it questionable that the author selects as a proxy for flood the maximum daily flow rates from a catchment that has a concentration time of about 1 day (line 193), after which they conclude that the main type of flood generation mechanism is precipitation. This seems to follow from the design of the study rather than a finding. The delayed response is hence more likely a result of the operation of the dams in the catchment.

There may be value in the manuscript that I overlook. But in its present form I cannot recommend publication, nor can I give good guidance on how the authors should improve the paper.

---

## Author Comment (AC1) · 28 May 2019

The authors would like to warmly thank the anonymous reviewer for his.her useful comments and suggestions which will help to significantly improve the manuscript. Please find below our answers to the main reviewer comments:

\*) It is unclear what we really learn. For example, the finding, which report correlations between floods and precipitation characteristics, these findings are only related based on the correlation of some floods of some catchment with some arbitrarily chosen P characteristics. This is potentially interesting, but does not answer any clear question to me, nor does it support the claim that "our results open new perspectives for flood

hazard assessments directly based on climate model outputs.". Sure rainfall is important in driving this floods, but this seems trivial, and I would expect some clear insight beyond this trivial statement. The provided topology of floods is an interesting way to summarize the type of flood events, but I do not see how it is really relevant beyond the case provided here.

Response:

According to your feedbacks as well as those from the 2 other reviewers, we realize that the objectives were probably not stated clearly enough in the first version of the manuscript. Please find below the clarified question and objectives.

A set of previous studies based on data series starting from the 1960s have shown that regular alpine floods (with annual/sub-annual occurrence) are complex events resulting from numerous processes in interaction like rain variability, snow/ice-melt dynamics, and soil moisture evolution (e.g. Merz and Blöschl, 2003; Sikorska et al., 2015; Brunner et al., 2017; Keller et al., 2018). On another hand, a set of studies focused on single flood cases seen as the largest historical floods and showed that these "extreme" floods mainly result from heavy precipitation accumulations (e.g. Blöschl et al., 2013; Ruiz-Bellet et al., 2015; Brönniman et al., 2018; Stucki et al., 2018).

As stated by Alfieri et al. (2015), "the assessments of the future flood hazard are commonly performed by coupling atmospheric climate projections with land-surface schemes and hydrological models". Accurate flood hazard projections are required by the decision makers in charge of flood risk reduction and water resources management at local to regional scales (Kundzewicz et al., 2016). However, expected changes in the magnitude and frequency of floods are highly uncertain, mainly due to i) the large uncertainties of extreme precipitation projections by the global and regional climate models (Sillmann et al., 2013; Kundzewicz et al., 2014; Mehran et al., 2014) and ii) the uncertainties of hydrological modelling (Dankers et al., 2014).

To overcome the uncertainties in the high-magnitude floods hazard projections, and

as proposed by Farnham et al. (2018), a complementary approach that would rely on direct links between atmospheric processes and flood occurrences is used in this study. This approach assumes that i) flood events mainly result from "extreme" precipitation and ii) that atmospheric features resulting in "extreme" precipitation can be used as predictors of such events directly from climate projections (e.g. Farnham et al., 2018; Schlef et al., 2019). In this study, we explore the first point, i.e. in what extent the generation of high-magnitude flood events in a large mountainous catchment can be explained by precipitation only. We also analyse the features of precipitation, i.e. its duration and its accumulation, associated with such natural hazard.

To reach this objective, we propose a new approach, at the intersection between the study of regular alpine floods and of largest historical floods, discussed above. We study historical floods that occurred in a given large mountainous catchment and we use long discharge and precipitation datasets (almost a century) to get a "robust" sample of high-magnitude flood events.

Our key results are:

- Precipitation alone seems sufficient to explain 13 of 28 flood events (types 2 and 4). Conversely, precipitation alone is not sufficient to explain the onset of flooding of types 1 and 3, possibly associated with other processes such as snow or ice melting.

- The largest flood events (return time period > 20 years) clearly result from precipitation accumulations only.

- Precipitation accumulations resulting in these flood events are characterized mostly by the 2-day and secondly by the 8-day accumulation, all ending 1 day before the events

- In this given catchment, only flood events with return time period > 20 years or types 2 and 4 flood events could be associated with atmospheric features. - To link these flood events to atmospheric features, a link between atmospheric processes and 2 and

8-day precipitation accumulations.

We achieve promising results since part (13 of 28 flood events) of the high-magnitude floods seem mainly associated with "extreme" precipitation accumulations only. Interestingly this includes the strongest flood events (return period > 20 years) that have the potential of greatest impacts on societies. Hence, this opens a promising avenue for complementary flood hazard projections if robust links can now be found between atmospheric processes and 2 and 8-day precipitation accumulations.

Since this approach mainly relies on the global gridded ERA-20C reanalysis, it can be applied in any part of the world. The main limitation is the need of a long flow series to get a large sample of high-magnitude flood events. A second limitation may relies on the need of meteorological station data to evaluate the precipitation series from the ERA-20C since they might encompass large biases (as suggested by the reviewer). We trust that this approach could be successfully applied in many parts of the world since we have shown that it can work for high-magnitude events in a mountainous catchment, where the flood-induced hydrometeorological processes are made even more complex by the topography, the presence of snow and ice, etc.

Alfieri, A., Burek, P., Feyen, L., Forzieri, G.: Global warming increases the frequency of river floods in Europe, hydrology and earth system sciences, 19, 2247-2260, doi:10.5194/hess-19-2247-2015, 2015.

Blöschl, G., Nester, T., Komma, J., Parajka, J., Perdigão, R.A.P.: The June 2013 flood in the Upper Danube Basin, and comparisons with the 2002, 1954 and 1899 floods, Hydrol. Earth Syst. Sci., 17, 5197–5212, doi:10.5194/hess-17-5197-2013, 2013.

Brönnimann, S., Rohr, C., Stucki, P., Summermatter, S., Bandhauer, M., Barton, Y., Fischer, A., Froidevaux, P., Germann, U., Grosjean, M., Hupfer, F., Ingold, K., Isotta, F., Keiler, M., Martius, O., Messmer, M., Mülchi, R., Panziera, L., Pfister, L., Raible, C. C., Reist, T., Rössler, O., Röthlisberger, V., Scherrer, S., Weingartner, R., Zappa, M., Zimmermann, M., Zischg, A. P.: 1868 – Les inondations qui changèrent la Suisse

: Causes, conséquences et leçons pour le futur, Geographica Bernensia, G94, 52 S., doi :10.4480/GB2018.G94.03, 2018.

Brunner, M.I., Viviroli, D., Sikorska, A.E., Vannier, O., Favre, A.C., Seibert, J.: Flood type specific construction of synthetic design hydrographs, Water Resour. Res, 53, 1390-1406, https://doi.org/10.1002/2016WR019535, 2017.

Dankers, R., Arnell, N.W., Clark, D.B., Falloon, P.D., Fekete, B.M., Gosling, S.N., Heinke, J., Kim, H., Masaki, Y., Satoh, Y., Stacke, T., Wada, Y., Wisser, D.: First look at changes in flood hazard in the Inter-Sectoral Impact Model Intercomparison Project ensemble, Proceedings of the National Academy of Sciences of the United States of America, 111, 3257-3261, https://doi.org/10.1073/pnas.1302078110, 2014.

Farnham, D.J., Doss-Gollin, J., Lall, U.: Regional extreme precipitation events: robust inference from credibly simulated GCM variables, water resource research, 54, 3809-3824, https://doi.org/10.1002/2017WR021318, 2018.

Keller, L., Rössler, O., Martius, O., Weingartner, R.: Delineation of flood generating processes and their hydrological response, Hydrological Processes, 32, 228-240, https://doi.org/10.1002/hyp.11407, 2018.

Kundzewicz, Z.W., Krysanova, V., Dankers, R., Hirabayashi, Y., Kanae, S., Hattermann, F.F., Huang, S., Milly, P.C.D., Stoffel, M., Driessen, P.P.J., Matczak, P., Quevauviller, P., Schellnhuber, H.J.: Differences in flood hazard projections in Europe – their causes and consequences for decision making, hydrological sciences journal, 62, 1-14, https://doi.org/10.1080/02626667.2016.1241398, 2016.

Kundzewicz, Z.W., Kanae, S., Seneviratne, S.I., Handmer, J., Nicholls, N., Peduzzi, P., Mechler, R., Bouwer, L.M., Arnell, N., Mach, K., Muir-wood, R., Brakenridge, G.R., Kron, W., Benito, G., Honda, Y., Takahashi, K., Sherstyukov, B.: Flood risk and climate change: global and regional perspectives, Hydrological Sciences Journal, 59, 1-28, https://doi.org/10.1080/02626667.2013.857411, 2014.

Mehran A., AghaKouchak A., Phillips T.J.: Evaluation of CMIP5 continental precipitation simulations relative to satellite-based gauge-adjusted observations. J. Geophys. Res. Atmos, 119, 1695-1707, 2014.

Merz, R. and Blöschl, G.: Regional flood risk — what are the driving processes?, Water resources research, 39 (12), 1340, doi:10.1029/2002WR001952, 2003.

Ruiz-Bellet, J.L., Balasch, J.C., Tuset, J., Barriendos, M., Mazon, J., Pino, D.: Historical, hydraulic, hydrological and meteorological reconstruction of 1874 Santa Tecla flash floods in Catalonia (NE Iberian Peninsula), Journal of Hydrology, 524, 279-295, http://dx.doi.org/10.1016/j.jhydrol.2015.02.023, 2015.

Schlef, K.E., Moradkhani, H., Lall, U.: Atmospheric circulation patterns associated with extreme United States floods indentified via machine learning, scientific reports, 9, first online published 9 may 2019, https://doi.org/10.1038/s41598-019-43496-w, 2019.

Sikorska, A.E., Viviroli, D., Seibert, J.: Flood-type classification in mountainous catchments using crisp and fuzzy decision trees, Water Resour. Res, 51, 7959-7976, https://doi.org/10.1002/2015WR017326, 2015.

Sillmann J., Kharin V.V., Zhang X., Zwiers F.W., Bronaugh D.: Climate extremes indices in the CMIP5 multimodel ensemble: Part 1. Model evaluation in the present climate. J. Geophys. Res. Atmospheres,118, 1716-1733, 2013.

Stucki, P., Bandhauer, M., Heikkilä, U., Rössler, O., Zappa, M., Pfister, L., Salvisberg, M., Froidevaux, P., Martius, O., Panziera, L., Brönnimann, S.: Reconstruction and simulation of an extreme flood event in the Lago Maggiore catchment in 1868, Nat. Hazards Earth Syst. Sci., 18, 2717-2739, https://doi.org/10.5194/nhess-18-2717-2018, 2018.

*) I am unconvinced the ERA data is accurate enough to do any kind of the inferences that are performed in this study. While it is argued that the data is accurate enough (to use percentiles), as supported by Fig A2, this figure to me shows a lot of scattering

which lead to the conclusion that these data are in fact highly unreliable to use. *)L173-181: You state that "[: : :], daily precipitation percentiles from the two datasets are in a good agreement (see Fig. A2 in the Appendix); when a high percentile value of precipitation accumulation is observed for a given day in one of the datasets, a high percentile value is also observed for this day in the other dataset.". However, when I look at Figure A2 I find this difficult to conclude with so much scatter.

Response:

We fully agree that the gridded ERA-20C reanalysis is not perfect in terms of daily precipitation. In Fig.A1, ERA-20C tends to underestimate the daily precipitation in comparison with the observations, especially for the extreme values.

However, for this study, we consider that the use of the percentile value (instead of the daily accumulation) limits the impact of the weakness of ERA-20C. Indeed, as shown in the appendix section B, the flood type classification remains the same whatever the used precipitation dataset (i.e. meteorological stations or ERA-20C reanalysis). This supports the use of percentile values of the daily precipitation series from the ERA-20C dataset back to 1923, i.e. the beginning of the flow data.

*) Why is soil moisture not considered in explaining the floods? Could this be ruled out a-priori for some reason?

Response:

In this study, the main objective is to explore in what extent the precipitation, only, may explain the high-magnitude flood events. Therefore, the use of other observations (e.g. glacier mass balance to document snow-melt) are not considered. Moreover, in this region, long-term observation of soil-moisture is rare. Furthermore, Norbiato et al. (2008) showed that the impact of the initial moisture conditions in runoff is negligible for large catchments with high storage capacity. Froidevaux et al. (2015) show that Pre-Alpine and Alpine catchments have a short discharge memory because of the weak

role of the precursor antecedent precipitation on the flood triggering.

Froidevaux, P., Schwanbeck, J., Weingartner, R., Chevalier, C., Martius, O.: Flood triggering in Switzerland: the role of daily to monthly preceding precipitation, Hydrol. Earth Syst. Sci., 19, 3903-3924, doi:10.5194/hess-19-3903-2015, 2015.

Norbiato, D., Borga, M., Degli Esposti, S., Gaume, E., Anquetin, S.: Flash flood warning based on rainfall thresholds and soil moisture conditions: An assessment for gauged and ungauges basins, Journal of hydrology, 362, 274-290, https://doi.org/10.1016/j.jhydrol.2008.08.023, 2008.

*)L36-37: Maybe it is my limitation, but I have no idea what a "socio-ecosystem" really refers too. Can you not just replace this by a less fancy sounding, but more straightforward word?

Response:

We fully understand that this term may not be familiar for some disciplines. This work is part of the "Cross Disciplinary Program Trajectories " that aims to improving knowledge on the interactions between human societies and their environment in the Alpine regions. The concept of "socio-ecosystem" is at the core of the project and is thus used by all the disciplines to favour the interdisciplinary approach. As mentioned by Gilberto Gallopin (1994) Âń A socio-ecological system refers to any system composed of a societal (or human) component and an ecological (or biophysical) component. Socio-ecological systems may be urban as well as rural. [. . .] Socio-ecological systems exist at various levels, ranging from the local (a household in interaction with its surroundings) to the global (consisting of all mankind and the ecosphere). Âż

Gallopin G.: Impoverishment and sustainable development, A report of the International Institut for Sustainable Development (IISD) – 1994. p.19.

*)L137-142: It is unclear to me why this normalization has been performed. i.e. what do you really mean by "To reduce the influence of the marked glacio-nival or nival regime

in the analysis of the discharges"? And why would you want to do that?

Response:

As shown in the Fig.1 below, the discharge series of Rhône@Bognes and of the three sub-catchments are affected by a marked seasonality.

Figs. 5 and 7 present the anomalies of discharge associated with the flood type (from D-7 to D+3). By using the deseasonalised anomalies, we obtained the "real" anomalies of the discharges, with respect to the season in which they occur. For a river affected by a strong seasonality as Rhône@Bognes is, the classical anomalies computed as the difference of the discharge with the mean annual discharge will be less representative of the "real" discharge anomalies values associated with the floods. The anomalies values depend on the seasonal mean base level of the river (here mainly higher during summer and lower during winter).

*)L186-189: I do not understand the description and meaning of Figs 2 and 3.

Response:

Figs 2 and 3 are indeed a bit difficult to understand, we therefore need to be clearer in the text to better guide the reader.

To capture the main flood characteristics (e.g. "short-rain" or "long-rain" floods from Merz and Blöschl, 2003), we tested different time sequences of precipitation occurring prior to the floods. This allows to highlight the time sequences that have the main influence on the 28 observed flood events.

We focus on two aspects: the precipitation duration (number of consecutive days) and the occurrence of the precipitation sequence (with respect to the flood day). In total, we studied 10 precipitation durations (from 1 to 10 consecutive days). The mean percentile is thus computed for these 10 durations. Then, to analyse the signature of the occurrence of the precipitation sequence on the flood day, we compute the mean percentile of the 10 precipitation sequences for 11 ending times (from 10 to 0 days prior

to the flood).

To illustrate this approach, Fig. 2 is proposed as a conceptual graphic illustrating the 10 precipitation sequences, i.e. from 1 to 10 consecutive days (y-axis) which end from 10 days to 0 day before the flood day (x-axis).

Fig.3 displays the mean percentile values associated with each of the precipitation duration (colours) and for each ending time (Up to...). Fig.3 aims at identifying the durations that have the greatest influence on the 28 flood events. The higher the percentile value is, the more the precipitation sequence is related to high precipitation accumulation, compared to the entire period studied. From Fig.3, the following comments can be made: the precipitation accumulation has the greatest influence on the floods when we consider the precipitation durations until the day before the flood whatever the precipitation durations. Therefore, for the rest of the study, the 10 precipitation durations that end one day before the flood events (up to D-1) are kept and analysed to then select which durations are the more informative.

Merz, R. and Blöschl, G.: Regional flood risk — what are the driving processes?, Water resources research, 39 (12), 1340, doi:10.1029/2002WR001952, 2003.

**Fig. 1.** hydrograms of the Rhône@Bognes, Rhône@HDI, Arve@BDM and for Valserine catchment for the 1923-2010 period.

---

## Author Comment (AC2) · 28 May 2019

The authors would like to warmly thank the anonymous reviewer for his.her useful comments and suggestions which will help to significantly improve the manuscript. Please find below our answers to the reviewer comments:

\*) The main findings could be a bit more highlighted and generalized.

response:

According to your feedbacks as well as those from the 2 other reviewers, we realize that the objectives were probably not stated clearly enough in the first version of the

manuscript. Please find in the response to the reviewer 1 the clarified question and objectives.

This paper focus on the development of a new approach that aims at establishing the links between atmospheric processes and flood occurrences (e.g. Farnham et al., 2018; Schlef et al., 2019). The main objective is to overcome the uncertainties in the high-magnitude floods hazard projections. This approach assumes that i) flood events mainly result from "extreme" precipitation and ii) that atmospheric features associated with such "extreme" precipitation can be used as predictors of these events directly from climate projections. In this study, we explore the first point, i.e. in what extent the generation of high-magnitude flood events in a large mountainous catchment can be explained by precipitation only.

Our key results are:

- Precipitation alone seems sufficient to explain 13 of 28 flood events (types 2 and 4). Conversely, precipitation alone is not sufficient to explain the onset of flooding of types 1 and 3, possibly associated with other processes such as snow or ice melting.

- The largest flood events (return time period > 20 years) clearly result from precipitation accumulations only.

- Precipitation accumulations resulting in these flood events are characterized mostly by the 2-day and secondly by the 8-day accumulation, all ending 1 day before the events

-In this given catchment, only flood events with return time period > 20 years or types 2 and 4 flood events could be associated to atmospheric features. -To link these flood events to atmospheric features, a link between atmospheric processes and 2 and 8-day precipitation accumulations.

We achieve promising results since part (13 of 28 flood events) of the high-magnitude floods seem mainly associated with "extreme" precipitation accumulations only. Inter-

estingly this includes the strongest flood events (return period > 20 years) that have the potential of greatest impacts on societies. Hence, this opens a promising avenue for complementary flood hazard projections if robust links can now be found between atmospheric processes and 2 and 8-day precipitation accumulations.

Since this approach mainly relies on the global gridded ERA-20C reanalysis, it can be applied in any part of the world. The main limitation is the need of a long flow series to get a large sample of high-magnitude flood events. A second limitation may relies on the need of meteorological station data to evaluate the precipitation series from the ERA-20C since they might encompass large biases (as suggested by the reviewers). We trust that this approach could be successfully applied in many parts of the world since we have shown that it can work for high-magnitude events in a mountainous catchment, where the flood-induced hydrometeorological processes are made even more complex by the topography, the presence of snow and ice, etc.

Farnham, D.J., Doss-Gollin, J., Lall, U.: Regional extreme precipitation events: robust inference from credibly simulated GCM variables, water resource research, 54, 3809-3824, https://doi.org/10.1002/2017WR021318, 2018.

Schlef, K.E., Moradkhani, H., Lall, U.: Atmospheric circulation patterns associated with extreme United States floods indentified via machine learning, scientific reports, 9, first online published 9 may 2019, https://doi.org/10.1038/s41598-019-43496-w, 2019.

*)It would really be interesting to look at the precipitation patterns leading to the superimposition of the flood peaks from the three subcatchments. The topology of the river network and the buffering effect of Lake Geneva suggests that the precipitation patterns (spatio-temporal distribtion of rainfall in the three subcatchments) leading to superimpostion of flood peaks and thus to high-magnitude floods must follow a certain pattern. I suggest to elaborate more on that in the discussion when introducing the "type 1 floods" (see, e.g., Pattison, I., Lane, S. N., Hardy, R. J., and Reaney, S. M.: The role of tributary relative timing and sequencing in controlling large floods, Water Re-

sour. Res., 5444–5458, doi:10.1002/2013WR014067, 2014. or Zischg, A. P., Felder, G., Weingartner, R., Quinn, N., Coxon, G., Neal, J., Freer, J., and Bates, P.: Effects of variability in probable maximum precipitation patterns on flood losses, Hydrol. Earth Syst. Sci., 22, 2759–2773, doi:10.5194/hess-22-2759-2018, 2018).

response:

As you said, such extensive study might be very interesting to carry out but it is slightly apart of our objectives; moreover, it will require high-resolution datasets (hourly time steps, few km2) that we do not currently have.

A first analysis of the rainfall and discharge co-evolutions is proposed in Fig. 7c. As a first finding, we observe a good agreement at the daily time scale between the evolutions of the flood peaks and the one-day precipitation sequences for the Arve and Valserine sub-catchments (the Geneva catchment never participate to the flood peak, due to the buffering of the Lake Geneva). The sub-catchment variability of the one-day precipitation sequences is thus probably weak as illustrated in Fig. 7a. However, the resolution of the ERA-20C precipitation series does not allow a more precise study of these processes.

Minor remarks:

*)line 341: In my opinion, the statement on the role of snowmelting cannot be concluded from the present study. The effect of snowmelt was not analysed; line 371: As above, the role of snowmelt, although as mixed process, cannot be stated without having analysed it in detail; line 394: same as above.

response:

We agree that without the use of any specific cryospheric observations, it is impossible to affirm the role of snowmelt in the flood types 3. It is an hypothesis. The manuscript will be changed to avoid the confusion.

*)lines 408-410: Please describe what you mean with "new perspectives".

response:

We agree with you that the sentence was a bit confusing. By "new perspectives", we meant that highlighting the predominant role of extreme precipitation for certain types of floods may help to explore the future trends of these high-impact floods. As explained in the response to the reviewer 1, since we showed that flood-types 2 and 4 may be explained by the precipitation alone, the next step is dedicated to the identification of the main associated atmospheric processes. Thus, we will use such proxies to explore the future occurrence of these precipitations and then the future occurrence of high-magnitude floods.

---

## Author Comment (AC3) · 28 May 2019

The authors would like to warmly thank the anonymous reviewer for his.her useful comments and suggestions which will help to significantly improve the manuscript. Please find below our answers to the reviewer comments:

*)The manuscript is difficult to follow. The flow data are first described as coming from three catchments, then two, and finally one catchment, being runoff from a heavily regulated catchment where the considered flow is the combined flow from two of the smallest catchments.
response:

Sorry for the confusion probably due to the lack of details when using the discharge dataset. The study focuses on the upper Rhône River catchment, for which the Rhône@Bognes gauge station records daily mean discharges at the outlet of the catchment (10 900 km2). To study the flood dynamic within this catchment, three sub-catchments have been also considered: (i) the Geneva catchment (8 000 km2) with the Rhône@HDI gauge station at the outlet of this sub-catchment, (ii) the Arve catchment (1 900 km2) with the Arve@BDM gauge station at the outlet of this sub-catchment, and (iii) the Valserine catchment (1 000 km2). For this latter sub-catchment, no gauge station is available, the daily mean discharge is thus estimated as follow: discharge of the Valserine catchment = Rhône@Bognes – (Rhône@HDI + Arve@BDM).

*)The data are described as being adjusted for seasonal variation, but all results are reported as flow values.

response:

In fact, we display deseasonalised anomalies evolutions of the discharge for the 4 flood-types in Figs. 5, 7 and B1, to capture how abnormal were the discharges during the floods. In this study, when we talk about "anomalies" of discharge, it always refers to the "deseasonalised anomalies".

However, we also show the daily flow values associated with the 28 flood events in the Fig. 6, this is not "anomaly" but the observed daily mean discharge values associated with the 28 floods, to identify the strongest floods of our sample of high-magnitude flood events.

The manuscript will be changed to avoid the confusion; we will clarify this point from the beginning.

*)With respect to the precipitation two sources are available, station data and ERA-20. I have not worked with ERA-20, but I expect that it - like the other ERA reanalysis

products - to some extend are based on measured station data for precipitation, at least for some of the years. Hence the study most likely uses and compares different data products on which on is based on the other. Please clarify what data sources are used in the study and how they are used.

response:

The gridded ERA-20C reanalysis are indeed used as daily precipitation dataset over the 1923-2010 period. The full description of the ERA-20C reanalysis is given in Poli et al. (2016), where the authors explained the data assimilation procedure. The observations assimilated in the ERA-20C reanalysis are:

- the marine wind;

- the surface pressure;

- the sea ice concentration;

- the sea surface temperature;

- the solar radiation;

- the tropospheric and stratospheric aerosols;

- ozone;

- and the greenhouse gases.

No precipitation observation is assimilated in the ERAC-20C building process. Hence, the ERA-20C precipitation dataset is independent from the weather station data we used for the evaluation.

Poli, P., Hersbach, H., Dee, D.P., Berrisford, P., Simmons, A.J., Vitart, F., Laloyaux, P., Tan, D.G.H, Peupley, C., Thépaut, J.N., Trémolet, Y., Holm, E.V., Bonavita, M., Isaksen, L., Fischer, M.: ERA-20C: An Atmospheric Reanalysis of the Twentieth Century, J. Clim., 29, 4083-4097, https://doi.org/10.1175/JCLI-D-15-0556.1, 2016.

*)I don't understand how Figure 2 can be used to deduct exactly which data are used in the cluster analysis and how.

response:

This point has been also highlighted by Reviewer 1. The text will be changed in order to better describe our methodology and to avoid the confusion.

Fig.2 is indeed a bit difficult to understand.

To capture the main flood characteristics (i.e. as proposed by Merz and Blöschl, 2003, short-rain or long-rain floods), we tested different time sequences of precipitation occurring prior to the floods. This allows to highlight the time sequences that have the main influence on the 28 observed flood events.

We focus on two aspects: the precipitation duration (number of consecutive days) and the occurrence of the precipitation sequence (with respect to the flood day). In total, we studied 10 precipitation durations (from 1 to 10 consecutive days). The mean percentile is thus computed for these 10 durations. Then, to analyse the signature of the occurrence of the precipitation sequence on the flood day, we compute the mean percentile of the 10 precipitation sequences for 11 ending times (from 10 to 0 days prior to the flood).

To illustrate this approach, Fig. 2 is proposed as a conceptual graphic illustrating the 10 precipitation sequences, i.e. from 1 to 10 consecutive days (y-axis) which end from 10 days to 0 day before the flood day (x-axis).

The selections of the precipitation durations and the ending times are based on the analyses of Fig.3 and 4, respectively. Fig.3 displays the mean percentile values associated with each of the precipitation duration (colours) and for each ending time (Up to. . .). It aims at identifying the durations that have the greatest influence on the 28 flood events. The higher the percentile value is, the more the precipitation duration is related to high precipitation accumulation, compared to the entire period studied. From

Fig.3, the following comments can be made: the precipitation accumulation has the greatest influence on the floods when we consider the precipitation sequences until the day before the flood (D-1) whatever the duration.

Fig.4 displays the distribution of percentile values for the 10 precipitation sequences ending at D-1. The selection of the best duration related to the high precipitation accumulation is based on precipitation sequences that present the highest mean percentile values and a weakest dispersion within the 28 flood events. The 2-day and 8-day durations are thus selected based on Fig.4.

*)I also find it questionable that the author selects as a proxy for flood the maximum daily flow rates from a catchment that has a concentration time of about 1 day (line 193), after which they conclude that the main type of flood generation mechanism is precipitation. This seems to follow from the design of the study rather than a finding. The delayed response is hence more likely a result of the operation of the dams in the catchment.

response:

We are really sorry but we do not fully understand the meaning of your comment.

Since our introduction was not clear enough to present our objectives (see the last point below), there is probably misunderstanding on the basic assumption of this study.

*)There may be value in the manuscript that I overlook. But in its present form I cannot recommend publication, nor can I give good guidance on how the authors should improve the paper.

response:

According to your feedbacks as well as those from the 2 other reviewers, we realize that the objectives were probably not stated clearly enough in the first version of the manuscript. Please find below the clarified question, motivations and objectives of this study. In the revised manuscript, the introduction will be thoroughly modified to make

these points clearer.

A set of previous studies based on data series starting from the 1960s have shown that regular alpine floods (with annual/sub-annual occurrence) are complex events resulting from numerous processes in interaction like rain variability, snow/ice-melt dynamics, and soil moisture evolution (e.g. Merz and Blöschl, 2003; Sikorska et al., 2015; Brunner et al., 2017; Keller et al., 2018). On another hand, a set of studies focused on single flood cases seen as the largest historical floods and showed that these "extreme" floods mainly result from heavy precipitation accumulations (e.g. Blöschl et al., 2013; Ruiz-Bellet et al., 2015; Brönniman et al., 2018; Stucki et al., 2018).

As stated by Alfieri et al. (2015), "the assessments of the future flood hazard are commonly performed by coupling atmospheric climate projections with land-surface schemes and hydrological models". Accurate flood hazard projections are required by the decision makers in charge of flood risk reduction and water resources management at local to regional scales (Kundzewicz et al., 2016). However, expected changes in the magnitude and frequency of floods are highly uncertain, mainly due to i) the large uncertainties of extreme precipitation projections by the global and regional climate models (Sillmann et al., 2013; Kundzewicz et al., 2014; Mehran et al., 2014) and ii) the uncertainties of hydrological modelling (Dankers et al., 2014).

To overcome the uncertainties in the high-magnitude floods hazard projections, and as proposed by Farnham et al. (2018), a complementary approach that would rely on direct links between atmospheric processes and flood occurrences is used in this study. This approach assumes that i) flood events mainly result from "extreme" precipitation and ii) that atmospheric features resulting in "extreme" precipitation can be used as predictors of such events directly from climate projections (e.g. Farnham et al., 2018; Schlef et al., 2019). In this study, we explore the first point, i.e. in what extent the generation of high-magnitude flood events in a large mountainous catchment can be explained by precipitation only. We also analyse the features of precipitation, i.e. its duration and its accumulation, associated with such natural hazard.
To reach this objective, we propose a new approach, at the intersection between the study of regular alpine floods and of largest historical floods, discussed above. We study historical floods that occurred in a given large mountainous catchment and we use long discharge and precipitation datasets (almost a century) to get a "robust" sample of high-magnitude flood events.

Our key results are:

- Precipitation alone seems sufficient to explain 13 of 28 flood events (types 2 and 4). Conversely, precipitation alone is not sufficient to explain the onset of flooding of types 1 and 3, possibly associated with other processes such as snow or ice melting.

- The largest flood events (return time period > 20 years) clearly result from precipitation accumulations only.

- Precipitation accumulations resulting in these flood events are characterized mostly by the 2-day and secondly by the 8-day accumulation, all ending 1 day before the events.

- In this given catchment, only flood events with return time period > 20 years or types 2 and 4 flood events could be associated with atmospheric features. - To link these flood events to atmospheric features, a link between atmospheric processes and 2 and 8-day precipitation accumulations.

We achieve promising results since part (13 of 28 flood events) of the high-magnitude floods seem mainly associated with "extreme" precipitation accumulations only. Interestingly this includes the strongest flood events (return period > 20 years) that have the potential of greatest impacts on societies. Hence, this opens a promising avenue for complementary flood hazard projections if robust links can now be found between atmospheric processes and 2 and 8-day precipitation accumulations.

Since this approach mainly relies on the global gridded ERA-20C reanalysis, it can be applied in any part of the world. The main limitation is the need of a long flow series

to get a large sample of high-magnitude flood events. A second limitation may relies on the need of meteorological station data to evaluate the precipitation series from the ERA-20C since they might encompass large biases (as suggested by the reviewer). We trust that this approach could be successfully applied in many parts of the world since we have shown that it can work for high-magnitude events in a mountainous catchment, where the flood-induced hydrometeorological processes are made even more complex by the topography, the presence of snow and ice, etc.

Alfieri, A., Burek, P., Feyen, L., Forzieri, G.: Global warming increases the frequency of river floods in Europe, hydrology and earth system sciences, 19, 2247-2260, doi:10.5194/hess-19-2247-2015, 2015.

Blöschl, G., Nester, T., Komma, J., Parajka, J., Perdigão, R.A.P.: The June 2013 flood in the Upper Danube Basin, and comparisons with the 2002, 1954 and 1899 floods, Hydrol. Earth Syst. Sci., 17, 5197–5212, doi:10.5194/hess-17-5197-2013, 2013.

Brönnimann, S., Rohr, C., Stucki, P., Summermatter, S., Bandhauer, M., Barton, Y., Fischer, A., Froidevaux, P., Germann, U., Grosjean, M., Hupfer, F., Ingold, K., Isotta, F., Keiler, M., Martius, O., Messmer, M., Mülchi, R., Panziera, L., Pfister, L., Raible, C. C., Reist, T., Rössler, O., Röthlisberger, V., Scherrer, S., Weingartner, R., Zappa, M., Zimmermann, M., Zischg, A. P.: 1868 – Les inondations qui changèrent la Suisse : Causes, conséquences et leçons pour le futur, Geographica Bernensia, G94, 52 S., doi :10.4480/GB2018.G94.03, 2018.

Brunner, M.I., Viviroli, D., Sikorska, A.E., Vannier, O., Favre, A.C., Seibert, J.: Flood type specific construction of synthetic design hydrographs, Water Resour. Res, 53, 1390-1406, https://doi.org/10.1002/2016WR019535, 2017.

Dankers, R., Arnell, N.W., Clark, D.B., Falloon, P.D., Fekete, B.M., Gosling, S.N., Heinke, J., Kim, H., Masaki, Y., Satoh, Y., Stacke, T., Wada, Y., Wisser, D.: First look at changes in flood hazard in the Inter-Sectoral Impact Model Intercomparison Project ensemble, Proceedings of the National Academy of Sciences of the United States of

America, 111, 3257-3261, https://doi.org/10.1073/pnas.1302078110, 2014.

Farnham, D.J., Doss-Gollin, J., Lall, U.: Regional extreme precipitation events: robust inference from credibly simulated GCM variables, water resource research, 54, 3809-3824, https://doi.org/10.1002/2017WR021318, 2018.

Keller, L., Rössler, O., Martius, O., Weingartner, R.: Delineation of flood generating processes and their hydrological response, Hydrological Processes, 32, 228-240, https://doi.org/10.1002/hyp.11407, 2018.

Kundzewicz, Z.W., Krysanova, V., Dankers, R., Hirabayashi, Y., Kanae, S., Hattermann, F.F., Huang, S., Milly, P.C.D., Stoffel, M., Driessen, P.P.J., Matczak, P., Quevauviller, P., Schellnhuber, H.J.: Differences in flood hazard projections in Europe – their causes and consequences for decision making, hydrological sciences journal, 62, 1-14, https://doi.org/10.1080/02626667.2016.1241398, 2016.

Kundzewicz, Z.W., Kanae, S., Seneviratne, S.I., Handmer, J., Nicholls, N., Peduzzi, P., Mechler, R., Bouwer, L.M., Arnell, N., Mach, K., Muir-wood, R., Brakenridge, G.R., Kron, W., Benito, G., Honda, Y., Takahashi, K., Sherstyukov, B.: Flood risk and climate change: global and regional perspectives, Hydrological Sciences Journal, 59, 1-28, https://doi.org/10.1080/02626667.2013.857411, 2014.

Mehran A., AghaKouchak A., Phillips T.J.: Evaluation of CMIP5 continental precipitation simulations relative to satellite-based gauge-adjusted observations. J. Geophys. Res. Atmos, 119, 1695-1707, 2014.

Merz, R. and Blöschl, G.: Regional flood risk — what are the driving processes?, Water resources research, 39 (12), 1340, doi:10.1029/2002WR001952, 2003.

Ruiz-Bellet, J.L., Balasch, J.C., Tuset, J., Barriendos, M., Mazon, J., Pino, D.: Historical, hydraulic, hydrological and meteorological reconstruction of 1874 Santa Tecla flash floods in Catalonia (NE Iberian Peninsula), Journal of Hydrology, 524, 279-295, http://dx.doi.org/10.1016/j.jhydrol.2015.02.023, 2015.

Schlef, K.E., Moradkhani, H., Lall, U.: Atmospheric circulation patterns associated with extreme United States floods indentified via machine learning, scientific reports, 9, first online published 9 may 2019, https://doi.org/10.1038/s41598-019-43496-w, 2019.

Sikorska, A.E., Viviroli, D., Seibert, J.: Flood-type classification in mountainous catchments using crisp and fuzzy decision trees, Water Resour. Res, 51, 7959-7976, https://doi.org/10.1002/2015WR017326, 2015.

Sillmann J., Kharin V.V., Zhang X., Zwiers F.W., Bronaugh D.: Climate extremes indices in the CMIP5 multimodel ensemble: Part 1. Model evaluation in the present climate. J. Geophys. Res. Atmospheres,118, 1716-1733, 2013.

Stucki, P., Bandhauer, M., Heikkilä, U., Rössler, O., Zappa, M., Pfister, L., Salvisberg, M., Froidevaux, P., Martius, O., Panziera, L., Brönnimann, S.: Reconstruction and simulation of an extreme flood event in the Lago Maggiore catchment in 1868, Nat. Hazards Earth Syst. Sci., 18, 2717-2739, https://doi.org/10.5194/nhess-18-2717-2018, 2018.